# SE-GA: Memory-Augmented Self-Evolution for GUI Agents

**Shilong Jin** [1 2]  **Lanjun Wang** [2 †]  **Zhuosheng Zhang** [3]

## Abstract

Autonomous Graphical User Interface (GUI) agents often struggle with multi-step tasks due to constrained context windows and static policies that fail to adapt to dynamic environments. To address these limitations, this work proposes the Self-Evolving GUI Agent (SE-GA), a novel framework that integrates hierarchical memory structures with an iterative self-improvement mechanism. At the core of our approach is Test-Time Memory Extension (TTME), which facilitates long-term planning by dynamically retrieving episodic, semantic, and experiential memories to provide salient contexts during inference. To ensure continuous learning, we introduce Memory-Augmented Self-Evolution (MASE), which is a training pipeline that adopts the data collected by TTME to stabilize and enhance the agent's foundational policy. Extensive evaluations across both offline and online benchmarks demonstrate SE-GA achieves state-of-the-art performance, reaching success rates of 89.0% on ScreenSpot and 75.8% on the challenging AndroidControl-High dataset. Furthermore, significant improvements on the AndroidWorld benchmark highlight the superior generalization to dynamic environments. Open source code: https://github.com/jinshilong-dev/SE-GA

## 1. Introduction

Graphical User Interface (GUI) serves as a universal bridge connecting human intent to digital execution, acting as a vital medium for human-computer interaction across diverse scenarios such as mobile applications, websites, and desktop software (Nguyen et al., 2025; Zhou et al., 2025b; Hurst et al., 2024). Developing autonomous agents capable of effectively navigating these interfaces enables automated task execution, thereby significantly enhancing human productivity (Wang et al., 2025a; Ye et al., 2025; Gu et al., 2025). While recent studies in Visual-Language Models (VLMs) have accelerated progress in this field, developing truly autonomous agents capable of adapting to dynamic environments remains a formidable challenge (Huang et al., 2025; Lu et al., 2024). Unlike static visual tasks, GUI navigation typically involves partial observability and uncertainty, including operational delays and dynamic layout changes. In multi-step tasks, even a single misstep can lead to irreversible failure (Cheng et al., 2025; Kong et al., 2025). Despite improvements in reactive decision-making, the performance of GUI agents in long-horizon tasks remains severely limited. This study identifies two key challenges that hinder current progress as follows.

First, GUI navigation tasks in the real world are partially observable and historically dependent, where critical information may appear only in early steps but continues to influence decisions far into the future. However, most existing methods rely primarily on the current screenshot and a limited context window (Wang et al., 2025c; Lu et al., 2025b; Liu et al., 2025b; Zhang et al., 2025a), failing to maintain a precise record of the full interaction history and leverage critical information. Consequently, they are vulnerable to error accumulation, where early mistakes or forgotten contexts often lead to irreversible failures in multi-step tasks.

Second, GUI navigation tasks in the real world are rarely isolated, as they often manifest as variations or compositions of previously completed tasks, requiring the reuse of past successful strategies for completion. However, current agents typically operate with static policies trained on fixed datasets or rely on temporary retrieval without a unified memory organization (Wu et al., 2024; Yuan et al., 2025; Gou et al., 2024). This limitation makes it impossible for current agents to extract and learn successful experiences, thus preventing them from generalizing learned knowledge to dynamic environments. Therefore, current GUI agents lack a unified mechanism to encode explicit historical experiences into implicit policy parameters, limiting them to static execution rather than achieving continuous self-evolution.

To address these challenges, this study aims to develop a

†Corresponding author. [1]College of Intelligence and Computing, Tianjin University, Tianjin, China [2]School of New Media and Communication, Tianjin University, Tianjin, China [3]School of Computer Science, Shanghai Jiao Tong University, Shanghai, China. Correspondence to: Lanjun Wang <wanglanjun@tju.edu.cn>.

*Proceedings of the 43rd International Conference on Machine Learning*, Seoul, South Korea. PMLR 306, 2026. Copyright 2026 by the author(s).

memory-augmented GUI agent that can reuse past experiences, and refine policy through continuous interaction, thereby transforming the GUI agent from a static command executor into a dynamic learner. Unlike traditional approaches constrained by fixed datasets, the Self-Evolving GUI Agent (SE-GA) proposed by this study continuously refines its policy through interaction. Specifically, SE-GA consists of two components: (1) Test-Time Memory Extension (TTME), a hierarchical retrieval system that enables precise context management over extended horizons when executing long-horizon tasks; (2) Memory-Augmented Self-Evolution (MASE), a two-stage training framework to stabilize learning in high-variance GUI environments.

In detail, TTME maintains a hierarchical memory repository during task execution to enhance the capabilities of SE-GA to execute multi-step tasks. Inspired by human cognitive architectures, TTME constructs a hierarchical memory repository comprising three components: episodic memory for tracking immediate task progress, semantic memory for storing domain-general rules, and experiential memory for retrieving successful trajectories from similar historical tasks. This hierarchical memory design enables SE-GA to retrieve precise information ranging from recent trajectories to abstract domain knowledge, thereby informing long-term decision-making. Beyond static retrieval, TTME functions as a dynamic buffer that accumulates novel successful trajectories in real-time during inference, thereby enabling the agent to achieve online evolution without immediate retraining. To prevent memory saturation and achieve continuous learning, this study proposes the two-stage training framework MASE, which leverages the high-quality interaction data curated within the memory repository to enhance the foundational capabilities of VLMs. The proposed MASE framework can effectively encode non-parametric experience into the intrinsic policy of the model to achieve stable and efficient self-evolution.

Extensive experiments on various benchmarks demonstrate that SE-GA achieves superior success rates and exhibits strong generalization across different applications. The contributions are summarized as follows:

- We propose SE-GA, a memory-augmented framework for GUI agents that systematically organizes and exploits historical interaction data to improve the reliability of multi-step task execution.
- We design TTME, a hierarchical test-time memory mechanism that integrates episodic, semantic, and experiential memories, enabling the agent to retrieve both the context of recent interactions and relevant past experiences in a unified manner.
- We introduce MASE, a two-stage training pipeline that combines grounding supervision with self-collected experience, including a Hindsight Goal-Shifting strategy

for data construction and a stabilized training method for iterative improvement.
- Extensive experiments on multiple benchmarks show that SE-GA consistently improves success rates and robustness over baselines, especially on long-horizon and complex tasks.

## 2. Related Work

### 2.1. GUI Agents

Recent studies in VLMs have enabled autonomous agents capable of perceiving and interacting with GUIs (Chen et al., 2024b), typically by translating visual perception into executable instructions (Niu et al., 2024). Unlike early studies that treated GUI navigation as static screen parsing using separate modules (Yao et al., 2022; Deng et al., 2023), VLM-based agents integrate screen understanding and action prediction to perform zero-shot or few-shot tasks on mobile and desktop platforms (Wang et al., 2025b; Xu et al., 2025; Wang et al., 2024a). However, these methods rely heavily on instantaneous visual observations, leading to failures when critical information is occluded or during long-term tasks requiring temporal context. Furthermore, reliance on policies from static datasets limits their ability to adapt to dynamic layout changes or learn from feedback in real-time (Nguyen et al., 2025), severely restricting the generalization ability of GUI agents in dynamic environments.

### 2.2. Memory Mechanisms

To overcome the limitations of context windows and short-term observation, researchers have introduced various memory mechanisms into agent workflows (Zhang et al., 2025b). Standard approaches employ Retrieval-Augmented Generation (RAG) or vector databases to store and retrieve textual history (Deng et al., 2023; Cheng et al., 2024; Hu et al., 2025). Advanced studies like ShowUI (Wang & Liu, 2024; Lin et al., 2025; Lu et al., 2025a) introduce hierarchical memory structures to manage short-term and long-term contexts, enabling agents to remember user preferences or historical dialogues (Wang et al., 2024b; Wu et al., 2025). However, existing memory systems focus on textual semantic retrieval, which often proves insufficient for handling the spatial and structural complexity of GUI elements. Furthermore, they often fail to explicitly construct high-value strategies for reuse in similar tasks, forcing agents to repeatedly solve identical subproblems and reducing efficiency.

### 2.3. Self-Evolution Method

The training paradigm for GUI agents has evolved from behavior cloning to more sophisticated reinforcement learning. Early work primarily focused on SFT based on human demonstrations (Sun et al., 2025; Liu et al., 2025a). To fur-

ther enhance decision-making capabilities, recent research employs reinforcement learning algorithms, such as Group Relative Policy Optimization (GRPO), to align agent behavior with human intentions (Shen et al., 2025; Zhou et al., 2025a). Additionally, self-evolution techniques attempt iterative performance optimization using the inputs and outputs of the same model (Fang et al., 2025). Despite these advances, the sparse reward problem in multi-step tasks makes the stable training of GUI agents challenging (Guo et al., 2025; Evstafev, 2025). Over extended trajectories, a single error often causes rewards to vanish, hindering standard reinforcement learning algorithms from effective optimization (Lu et al., 2025b; Luo et al., 2025).

## 3. Problem Definition

GUI interaction presents unique challenges due to partial observability and uncertainties in system latency (Wang et al., 2025a). We formalize the GUI navigation task as a Partially Observable Markov Decision Process (POMDP), defined by the tuple $\langle \mathcal{S}, \mathcal{A}, \mathcal{O}, \mathcal{T}, \mathcal{R}, \gamma \rangle$, where $\mathcal{S}$ represents the set of environment states, $\mathcal{A}$ denotes the space of available actions, and $\mathcal{O}$ refers to the observation space. The transition function $\mathcal{T}(s_{t+1} \mid s_t, a_t)$ specifies the state transition probabilities, while the reward function $\mathcal{R}(s_t, a_t)$ provides the feedback signal. The discount factor $\gamma \in [0, 1]$ balances the weights between different types of rewards.

At each time step $t$, the agent receives a user instruction $Q$, an image observation $o_t \in \mathcal{O}$ from the GUI environment, and structured memory retrieved from the memory repository, denoted as $M_{\text{retrieved}}$. Specifically, the input $x_t$ received by the agent is defined as $x_t = (o_t, Q, M_{retrieved})$. After receiving the input, the agent generates an action $a_t$ through a structured reasoning process based on its policy, defined as $\pi_\theta(a_t | x_t)$. This process is augmented by a hierarchical memory context $\mathcal{M}_t$ to enhance reasoning capabilities, enabling the agent to make better decisions by incorporating past experiences. The agent then executes $a_t$, receives a new observation $o_{t+1}$, and a reward $r_t$ for this action, repeating this interaction loop until the instruction is completed or a terminal state is reached.

## 4. Methodology

As shown in Fig. 1, SE-GA consists of two components, TTME (Sec. 4.1) and MASE (Sec. 4.2), which together enable self-evolution in dynamic environments. This section describes these two components in detail.

### 4.1. Test-Time Memory Extension

To achieve reliable long-horizon decision-making in GUI navigation, the TTME module maintains a hierarchical memory repository $\mathcal{M} = (M^{EPI}, M^{SEM}, M^{EXP})$,

where $M^{EPI}$ stores executed historical actions, $M^{SEM}$ encodes general action rules, and $M^{EXP}$ contains high-value experiences distilled from previously completed tasks.

#### 4.1.1. EPISODIC MEMORY

When a GUI agent executes tasks, recording previously performed actions typically helps it better understand the current state and make accurate decisions. Therefore, we introduce an episodic memory repository $M^{EPI}$ to store historical actions during task execution. $M^{EPI}$ functions as a short-term working memory that tracks the actions taken to accomplish the current task. Formally, the episodic memory at time step $t$, denoted as $M_t^{EPI}$, is defined as follows:

$$M_t^{EPI} = [m_k]_{k=1}^{t-1}, \quad \text{where } m_k = \langle o_k, a_k, o_{k+1} \rangle. \quad (1)$$

However, maintaining the entire action history incurs unnecessary computational overhead and may introduce stale information that can mislead the agent into making erroneous decisions. Therefore, we employ a sliding-window mechanism with a fixed horizon $H$, retaining only the portion of the history that is most relevant to the current state. The episodic context at time step $t$, denoted as $\mathcal{C}_t^{epi}$, is constructed by retrieving transition subsequences that are strictly constrained within this time window. Consequently, $\mathcal{C}_t^{epi}$ summarizes the recent action trajectory of the agent:

$$\mathcal{C}_t^{epi} = [m_k]_{k=\epsilon}^{t-1}, \quad \text{where } \epsilon = \max(1, t - H). \quad (2)$$

This sliding-window truncation strategy keeps the input of GUI agents focused on recent relevant actions, while filtering out stale interaction history that may introduce irrelevant information.

#### 4.1.2. SEMANTIC MEMORY

While episodic memory effectively supports the GUI agent in making short-term decision-making, the agent also requires stable and generalizable knowledge to better understand the current state. A semantic memory repository $M^{SEM}$ is utilized to store abstract knowledge, such as universal interaction logic (e.g., "Log in before accessing restricted pages"). $M^{SEM}$ serves as a persistent long-term knowledge repository that accumulates universal rules to facilitate transfer across tasks. Specifically, for a task $i$, the repository consists a set of knowledge entries, where each entry $m_i^{sem}$ is defined as follows:

$$m_i^{sem} = \langle k_i^{sem}, d_i \rangle, \quad (3)$$

where $d_i$ denotes the textual description of the interaction rule, and $k_i^{sem} = \phi(Q_{hist})$ is its corresponding vector representation. To effectively retrieve relevant prior knowledge for the current task, we adopt an embedding-based similarity retrieval mechanism. Given the current user instruction

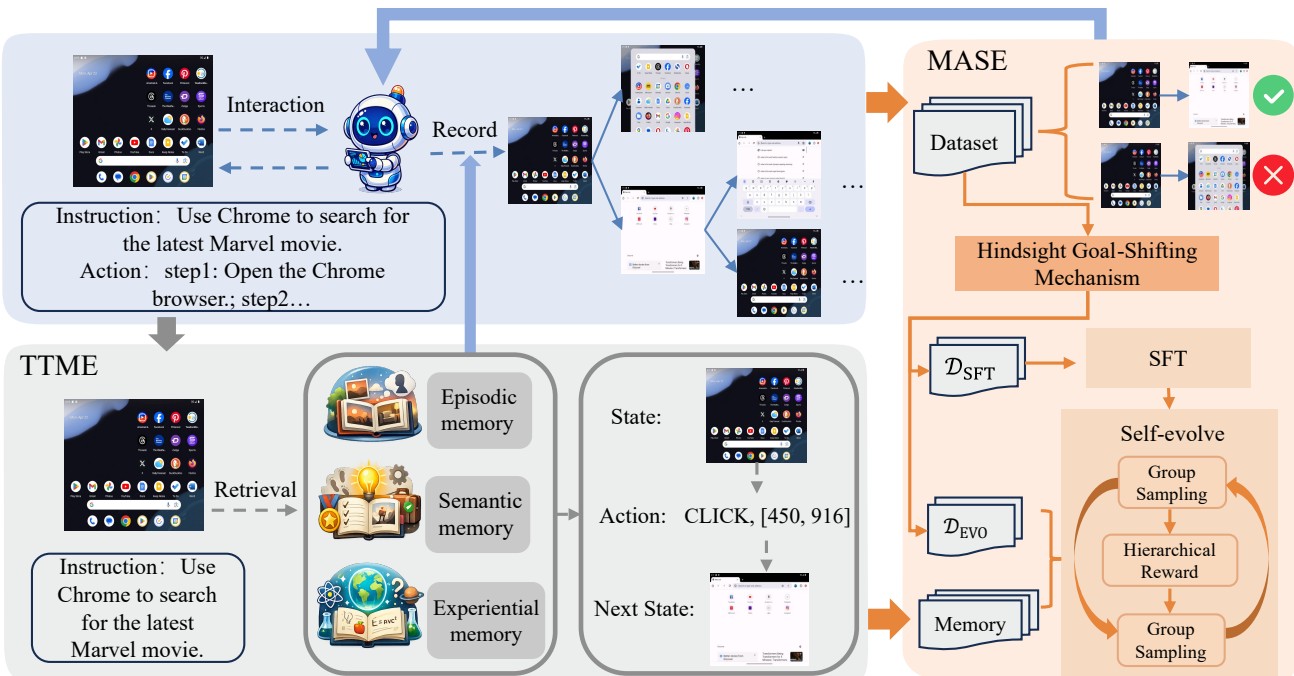

*Figure 1.* An overview of SE-GA, which can achieve self-evolution without relying on predefined tasks or human annotations. SE-GA begins with a model-free, interaction-driven traversal in online environments. This process uses TTME and generates a large number of triples consisting of actions and their corresponding pre-interaction and post-interaction screenshots, along with corresponding memories. Subsequently, SE-GA uses MASE to conduct self-evolution by using the collected data.

$Q$, the relevance score $S^{sem}$ of candidate entries $m_i^{sem}$ is computed using cosine similarity:

$$S^{sem}(Q, m_i^{sem}) = \frac{\phi(Q) \cdot k_i^{sem}}{|\phi(Q)||k_i^{sem}|}. \qquad (4)$$

The semantic context, denoted as $\mathcal{C}^{sem}$, is constructed by aggregating the descriptions of the Top-K entries with the highest relevance scores. This retrieval strategy provides the agent with general rules, enabling a better understanding of the behavioral logic underlying the current state.

### 4.1.3. EXPERIENTIAL MEMORY

Beyond episodic and semantic memory, experiences from previously executed similar tasks also provide valuable guidance. Therefore, we introduce an experiential memory repository $M^{EXP}$ to store such historical trajectories, improving the adaptability of the agent in dynamic environments. $M^{EXP}$ functions as a reference repository that allows the agent to recall and reuse past task execution strategies for decision-making. Specifically, each experiential entry $m_i^{exp}$ in the repository is defined as follows:

$$m_i^{exp} = \langle \tau_i, g(\tau_i), k_i^{intent}, k_i^{task} \rangle, \qquad (5)$$

where $\tau_i$ denotes the recorded raw trajectory, and $g(\tau_i)$ is a reflective summary synthesized by the agent. To support accurate retrieval across multiple modalities, we adopt a

hybrid retrieval mechanism that jointly considers semantic and visual features. Given the current user instruction $Q$ and the image observation $o_t$, the retrieval score $S^{exp}$ for candidate entries $m_i^{exp}$ is computed via a weighted fusion of intent consistency and visual similarity:

$$\begin{aligned} S^{exp}(Q, o_t) = &\lambda \cdot \text{Sim}(\phi(Q), k_i^{intent}) + \\ &(1 - \lambda) \cdot \text{Sim}(\psi(o_t), k_i^{task}), \end{aligned} \qquad (6)$$

where $\psi(\cdot)$ denotes the visual encoder, and $\lambda$ is a hyper-parameter that balances the contributions of semantic and visual features. The experiential context, denoted as $\mathcal{C}^{exp}$, is constructed by extracting the reflective summaries $g(\tau_i)$ from the Top-K entries with the highest scores. This retrieval strategy allows the agent to exploit past experiences when handling similar objectives or tasks.

Finally, the summaries in $\mathcal{C}^{exp}$ are incorporated into the input of the GUI agent, together with $\mathcal{C}^{epi}$ and $\mathcal{C}^{sem}$, providing guidance for reasoning at the current decision step.

### 4.2. Memory-Augmented Self-Evolution

To enable continuous learning and dynamic adaptation for GUI agents in real world, this study proposes a training framework termed MASE, which consists of two stages: *Grounding Training* and *Self-Evolution Training*.

### 4.2.1. STAGE I: GROUNDING TRAINING

GUI agents often struggle to translate high-level user instructions into low-level actions due to the gap between visual perception and the executable action space. To enhance the ability of the agent to analyze the current GUI state, integrate historical context, and infer an appropriate strategy, we employ supervised fine-tuning (SFT) to strengthen the reasoning capabilities of the model. Specifically, we formulate this process as a memory-aware behavior cloning task, optimizing the parameter set $\theta$ by minimizing the negative log-likelihood over the expert trajectories:

$$\mathcal{L}_{SFT}(\theta) = - \mathbb{E}_{(x,y) \sim \mathcal{D}_{ground}}$$
$$\left[ \frac{1}{|y|} \sum_{t=1}^{|y|} \log \pi_\theta(y_t \mid o_t, Q, M, y_{<t}) \right]. \quad (7)$$

### 4.2.2. STAGE II: SELF-EVOLUTION TRAINING

To further equip the agent with the ability to capture complex dependencies in GUI interactions, this study conducts model training based on GRPO (Guo et al., 2025) and introduces several targeted improvements. While GRPO aggregates advantages at the sequence level, GUI navigation tasks often involves critical intermediate steps where fine-grained credit assignment is essential. Therefore, we adopt a token-level importance ratio $\rho_{i,t}$, inspired by DAPO (Yu et al., 2025), to prevent irrelevant tokens from dominating the updates and inducing high-variance gradients. Specifically, for each context $x$, we sample a group of $G$ outputs $\{y_1, y_2, \ldots, y_G\}$ from the old policy $\pi_{\theta_{old}}$. The resulting optimization objective is formulated as follows:

$$\mathcal{J}(\theta) = \mathbb{E}_{x \sim \mathcal{D}_{evolve}} \left[ \frac{1}{\sum_{i=1}^{G} |y_i|} \sum_{i=1}^{G} \sum_{t=1}^{|y_i|} \right.$$
$$\left. \left( \min(\rho_{i,t} A_i, \rho_{i,t}^{clip} A_i) - \beta \mathbb{D}_{KL}(\pi_\theta || \pi_{ref}) \right) \right], \quad (8)$$

where $\pi_{ref}$ is the reference policy used to regularize the update via a KL-divergence constraint, thereby preventing mode collapse. $\rho_{i,t}$ denotes the token-level importance ratio, and $\rho_{i,t}^{clip}$ applies the adaptive clipping. $A_i$ represents the advantage computed from group-relative rewards. The resulting formulation is:

$$\rho_{i,t} = \frac{\pi_\theta(y_{i,t}|x, y_{i,<t})}{\pi_{\theta_{old}}(y_{i,t}|x, y_{i,<t})}, \quad A_i = \frac{r_i - \text{mean}(\{r_i\}_{i=1}^{G})}{\text{std}(\{r_i\}_{i=1}^{G})}. \quad (9)$$

**Adaptive Clipping.** To mitigate the issue that high-confidence correct tokens may be overly constrained, we introduce an adaptive upper clipping bound. This adaptive clipping design allows the model to make larger policy updates in the early stages of training, while progressively

tightening the constraint as training proceeds. In contrast to standard symmetric clipping, we define $\rho_{i,t}^{clip}$ using a dynamic upper threshold $\epsilon_{cur}$:

$$\rho_{i,t}^{clip} = \text{clip}\left(\rho_{i,t}, 1 - \epsilon, 1 + \epsilon_{cur}\right), \quad (10)$$

where $\epsilon_{cur}$ follows a cosine decay schedule with respect to the training progress $k/K$:

$$\epsilon_{cur} = \epsilon_{end} + \frac{1}{2}(\epsilon_{init} - \epsilon_{end})(1 + \cos(\pi \cdot \frac{k}{K})). \quad (11)$$

**Hierarchical Reward Design.** The design of the reward function $R(y, x)$ plays a critical role in guiding the agent to solve complex GUI tasks. We propose a hierarchical reward design method that evaluates model outputs from both format correctness and task execution accuracy:

$$R_{\text{total}} = w_f \cdot R_{\text{format}} + w_a \cdot R_{\text{acc}}, \quad (12)$$

where $R_{\text{format}}$ verifies whether the model output $y$ conforms to the expected format, returning 1 if valid and 0 otherwise. $R_{\text{acc}}$ measures content accuracy and is evaluated only when $R_{\text{format}} = 1$, ensuring the agent first learns to produce structurally valid outputs. $w_f$ and $w_a$ are weighting hyperparameters $w_f + w_a = 1$.

The accuracy reward $R_{\text{acc}}$ is customized according to specific task types. For evaluating sequences of GUI actions, this study provides fine-grained feedback by combining rewards for the action type and its parameters:

$$R_{\text{acc}} = w_t \cdot R_{\text{type}} + w_p \cdot R_{\text{param}}, \quad (13)$$

where $w_t + w_p = 1$. $R_{\text{type}}$ assigns a reward of 1 if the predicted action type (e.g., "click", "scroll") matches the ground truth, and 0 otherwise. Depending on the action category, $R_{\text{param}}$ is evaluated using two different criteria: *Grounding Task Rewards* and *Other Task Rewards*.

*Grounding Task Rewards*: For evaluating GUI element localization, this study adopts two evaluation strategies:

- Point Localization Reward ($R_{\text{point}}$): For the task type of click, given a predicted point coordinate $(x_p, y_p)$ and the ground-truth bounding box $B_{\text{gt}}$ of the target element, the reward is set to 1 if the predicted point lies inside the bounding box, and 0 otherwise:

$$R_{\text{param}} = R_{\text{point}} = \mathbb{I}((x_p, y_p) \in B_{\text{gt}}). \quad (14)$$

- Bounding Box Reward ($R_{\text{bbox}}$): For the task type of scroll, this study computes the Intersection over Union (IoU) between the predicted bounding box $B_{\text{pred}}$ and the ground-truth box $B_{\text{gt}}$. To avoid penalizing minor deviations excessively while encouraging significant overlap, we introduce a threshold $\tau_{\text{IoU}}$. The reward is

set to 1 if the IoU exceeds $\tau_{\text{IoU}}$; otherwise, it is defined as $\text{IoU}/\tau_{\text{IoU}}$.

$$R_{\text{param}} = R_{\text{bbox}}$$
$$= \begin{cases} 1 & \text{if } \text{IoU}(B_{\text{pred}}, B_{\text{gt}}) \geq \tau_{\text{IoU}} \\ \frac{\text{IoU}(B_{\text{pred}}, B_{\text{gt}})}{\tau_{\text{IoU}}} & \text{if } \text{IoU}(B_{\text{pred}}, B_{\text{gt}}) < \tau_{\text{IoU}} \end{cases} \tag{15}$$

*Other Task Rewards*: For other tasks (e.g., text input, numerical calculation), this study uses exact match or mathematical expression verification against the ground truth $y_{\text{gt}}$ to determine correctness:

$$R_{\text{param}} = R_{\text{other}}$$
$$= \mathbb{I}(\text{ExactMatch}(y_{\text{ans}}, y_{\text{gt}}) \vee \text{MathVerify}(y_{\text{ans}}, y_{\text{gt}})). \tag{16}$$

## 5. Experiments

In this section, we evaluate SE-GA on a diverse set of benchmarks designed to assess the capabilities of GUI agents. We adopt Qwen2.5-VL-7B (Bai et al., 2025) as the base model. Implementation details are provided in Sec. 5.1. Sec. 5.2 introduces a comprehensive overview of the evaluation benchmarks and metrics. Experimental results on each benchmark are reported in Sec. 5.3. Due to the absence of complete implementation details in prior work, this study references the results reported by UI-TARS (Wang et al., 2025a) to ensure a fair comparison. Ablation studies are provided in Sec. 5.4 to validate the effectiveness of each component.

### 5.1. Implementation Details

This study establishes a rigorous data construction pipeline to construct a comprehensive dataset containing 4K trajectories. Initially, we leverage several established open-source datasets as foundational sources, including AITW (Rawles et al., 2023), AMEX (Chai et al., 2025), and GUIOdyssey (Lu et al., 2025a). Subsequently, we apply Qwen-VL (Bai et al., 2025) to filter out overly simplistic or ambiguous samples, thereby curating a high-quality static subset. In addition, we collect new trajectories by interacting with an Android simulator and storing them in the memory repository. However, the resulting raw dataset inevitably contains a substantial number of invalid or low-quality trajectories, making it unsuitable for direct use. Inspired by retrospective experience replay mechanisms and hindsight experience replay (Mnih et al., 2013; Andrychowicz et al., 2018), we propose a novel data refinement method to further improve data quality. The detailed information of the dataset is provided in Appendix A.

**Hindsight Goal-Shifting Mechanism.** Given a failed trajectory $\tau = (s_0, a_0, s_1, \ldots, s_T)$ originally intended to

achieve goal $g$, if a prefix subsequence $\tau_{0:k}$ satisfies an alternative valid sub-goal $g'$ (e.g., the application is successfully opened but subsequent search operations fail), the trajectory is relabeled as a successful instance for $g'$. This process yields an expanded sample set $\mathcal{D}_{GS}$, which is then merged into $\mathcal{D}_{total}$, effectively converting failures into useful supervision signals for sub-task execution:

$$\mathcal{D}_{GS} = \{(\tau_{0:k}, g') \mid \text{Verify}(\tau_{0:k}, g') = 1, (\tau, g) \in \mathcal{D}_{collected}\}. \tag{17}$$

The final dataset $\mathcal{D}_{total}$ consists of two subsets: 2K samples $\mathcal{D}_{ground}$ for *Grounding Training* to maintain the fundamental capabilities of agents and to prevent catastrophic forgetting during the training process, and 2K samples $\mathcal{D}_{evolve}$ for *Self-Evolution Training*, which drives continuous improvement by learning from newly collected memories.

All experiments are conducted using 4 NVIDIA A800 GPUs. In the first training stage, the learning rate is set to 2e-6 with a global batch size of 16. In the second training stage, the learning rate is set to 2e-5, the global batch size to 256, and the group size is set to 16. Additionally, due to the absence of complete implementation details in prior work, this study references some experimental results reported by UI-TARS (Wang et al., 2025a) to ensure a fair comparison.

### 5.2. Evaluation Details

**Datasets.** Referring to previous work (Wang et al., 2025a), this study evaluates the performance of SE-GA on several benchmarks: (1) *ScreenSpot* (Cheng et al., 2024); (2) *AndroidControl* (Li et al., 2024); (3) *GUIOdyssey* (Lu et al., 2025a); (4) *AndroidWorld* (Rawles et al., 2024). More details about datasets are described in Appendix A.

**Metrics.** We employ a multi-dimensional evaluation protocol: (1) *Action Type Accuracy*, which measures the proportion of steps where the predicted action type matches the ground truth; (2) *Grounding Accuracy*, which evaluates spatial precision, counting a prediction as correct if the predicted coordinates fall inside the target bounding box or satisfy a predefined IoU threshold; (3) *Success Rate*, which reflects overall task completion.

**Baselines.** We compare SE-GA against fifteen recent baselines, grouped into three closed-source VLMs: GPT-4o (Hurst et al., 2024), Claude3-Opus (Anthropic, 2024), and Gemini-1.5-pro (Team et al., 2024); four generalist open-source VLMs: Qwen2.5-VL (7B and 72B) (Bai et al., 2025), InternVL2 (Chen et al., 2024a), Aria-UI (Yang et al., 2025); and eight specialized GUI agents: UI-TARS (7B and 72B) (Wang et al., 2025a), OS-Atlas (Wu et al., 2024), Aguvis (Xu et al., 2025), SeeClick (Cheng et al., 2024), UGround (Qian et al., 2025), OS-Genesis (Sun et al., 2025), and GUI-Critic-R1 (Wanyan et al., 2025).

*Table 1.* Performance comparison on ScreenSpot (Cheng et al., 2024). Best result is in **bold** and second-best result is in underline.

| Model | Model Size | Mobile | | Desktop | | Web | | Avg |
|---|---|---|---|---|---|---|---|---|
| | | Text | Icon | Text | Icon | Text | Icon | |
| GPT-4o | - | 20.2 | 24.9 | 21.1 | 23.6 | 12.2 | 7.8 | 18.3 |
| Claude | - | - | - | - | - | - | - | 83.0 |
| Gemini | - | - | - | - | - | - | - | 84.0 |
| UI-TARS | 72B | 94.9 | 82.5 | 89.7 | **88.6** | 88.7 | **85.0** | 88.4 |
| Qwen2.5VL | 72B | 95.0 | 80.0 | 95.3 | 74.3 | 87.4 | 69.4 | 85.0 |
| Aria-UI | 23B | 92.3 | 73.8 | 93.3 | 64.3 | 86.5 | 76.2 | 82.4 |
| SeeClick | 9.6B | 78.0 | 52.0 | 72.2 | 30.0 | 55.7 | 32.5 | 53.4 |
| Qwen2.5VL | 7B | 84.6 | 67.7 | 85.4 | 67.8 | 77.4 | 70.0 | 76.1 |
| UGround | 7B | 82.8 | 60.3 | 82.5 | 63.6 | 80.4 | 70.4 | 71.5 |
| OS-Atlas | 7B | 93.0 | 72.9 | 91.8 | 62.9 | 90.9 | 74.3 | 82.5 |
| Aguvis | 7B | 95.6 | 77.7 | 93.8 | 67.1 | 88.3 | 75.2 | 84.4 |
| SE-GA(Ours) | 7B | **96.3** | **83.0** | 95.9 | 76.4 | **91.0** | 84.0 | **89.0** |

*Table 2.* Performance comparison on AndroidControl (Li et al., 2024) and GUIOdyssey (Lu et al., 2025a). Best result is in **bold** and second-best result is in underline.

| Model | Model Size | AndroidControl-Low | | | AndroidControl-High | | | GUIOdyssey | | |
|---|---|---|---|---|---|---|---|---|---|---|
| | | Type | Grounding | SR | Type | Grounding | SR | Type | Grounding | SR |
| GPT-4o | - | 74.3 | 0.0 | 19.4 | 66.3 | 0.0 | 12.5 | 34.3 | 0.0 | 3.3 |
| Claude | - | 74.3 | 0.0 | 19.4 | 63.7 | 0.0 | 20.8 | 60.9 | 0.0 | 3.1 |
| UI-TARS | 72B | **98.1** | 89.9 | **91.3** | 83.7 | **81.5** | 74.7 | 95.4 | **91.4** | **88.6** |
| Aria-UI | 23B | - | 87.7 | 67.3 | - | 43.2 | 10.2 | - | 86.8 | 36.5 |
| SeeClick | 9.6B | 93.0 | 73.4 | 75.0 | 82.9 | 62.9 | 59.1 | 71.0 | 52.4 | 53.9 |
| InternVL-2 | 4B | 90.9 | 84.1 | 80.1 | 84.1 | 72.7 | 66.7 | 82.1 | 55.5 | 51.5 |
| OS-Atlas | 7B | 93.6 | 88.0 | 85.2 | **85.2** | 78.5 | 71.2 | 84.5 | 67.8 | 62.0 |
| Qwen2.5-VL | 7B | 83.4 | 87.0 | 65.5 | 68.6 | 59.7 | 47.0 | 55.6 | 37.7 | 34.3 |
| SE-GA(Ours) | 7B | 94.6 | **92.5** | 88.6 | 84.4 | 76.4 | **75.8** | 96.5 | 87.4 | 83.9 |

## 5.3. Main Results

### 5.3.1. GUI GROUNDING EVALUATION

**ScreenSpot.** Table 1 reports the grounding accuracy of SE-GA and baseline models on both text and icon elements. Notably, SE-GA achieves an average score of 89.0, consistently outperforming all 7B baselines and even surpassing larger models such as UI-TARS-72B and Qwen2.5-VL-72B. These gains can be attributed to the Hierarchical Reward Design in the MASE framework, particularly the explicit coordinate constraints ($R_{point}$ and $R_{bbox}$). By grounding visual perception in precise spatial feedback, SE-GA effectively mitigates the limitations of implicit vision-language alignment adopted by most baselines, which often struggle with pixel-level deviations in densely packed GUI layouts.

### 5.3.2. OFFLINE GUI AGENT EVALUATION

**AndroidControl.** Table 2 summarizes the step-level accuracy and success rates for SE-GA compared to the baselines on low-level execution and high-level planning tasks. On high-level tasks, SE-GA achieves a success rate of 75.8%, which surpasses all baseline methods with the same parameter scale and remains comparable in overall performance

*Table 3.* Performance comparison on AndroidWorld (Rawles et al., 2024). Best result is in **bold** and second-best result is in underline.

| Model | Model Size | AndroidWorld |
|---|---|---|
| GPT-4o | - | 23.7 |
| Qwen2.5-VL | 7B | 25.5 |
| OS-Genesis | 7B | 17.4 |
| GUI-Critic-R1 | 7B | 27.6 |
| UI-TARS | 7B | 33.0 |
| SE-GA(Ours) | 7B | **39.0** |

to the UI-TARS-72B model. These improvements are primarily attributed to the TTME module, particularly its hierarchical retrieval mechanism, which enables the agent to make decisions based on a coherent and structured history of past interactions. In contrast, baselines that rely on fixed context windows and implicit reasoning tend to lose critical information from earlier steps in long-horizon episodes, which ultimately leads to a decline in long-term planning performance and task success rates.

**GUIOdyssey.** Table 2 reports the step success rate and action type accuracy for SE-GA and the baselines on cross-app navigation tasks. Notably, SE-GA records a step success

*Table 4.* Ablation study of SE-GA components. "w/o TTME" indicates whether the hierarchical memory system is included, and "w/o MASE" indicates whether this study uses self-evolution training that improves the continuous learning ability of the GUI agent.

| Model | Model Size | AndroidControl-Low | | | AndroidControl-High | | | GUIOdyssey | | |
|---|---|---|---|---|---|---|---|---|---|---|
| | | Type | Grounding | SR | Type | Grounding | SR | Type | Grounding | SR |
| SE-GA | 7B | 94.6 | 92.5 | 88.6 | 84.4 | 76.4 | 73.8 | 96.5 | 87.4 | 83.9 |
| w/o TTME | 7B | 91.9 | 90.8 | 83.0 | 81.0 | 68.4 | 61.4 | 87.7 | 74.2 | 74.9 |
| w/o MASE | 7B | 88.6 | 86.9 | 74.3 | 72.2 | 60.1 | 59.7 | 84.3 | 52.2 | 60.4 |

rate of 83.9%, establishing a new state-of-the-art among 7B models and achieving the highest action type accuracy of 96.5% across all evaluated models, even outperforming UI-TARS-72B. This strong performance on both metrics indicates that SE-GA not only executes individual actions more accurately, but also maintains more reliable long-horizon decision-making across complex, multi-app workflows. These advancements can be attributed to the retrieval mechanism of TTME and the training paradigm of MASE. Specifically, MASE strengthens the foundational capabilities of the agent by learning from successful trajectories, while TTME facilitates decision-making in novel tasks by leveraging verified historical strategies. This dynamic approach is more robust than the static policy weights of standard baselines, which are inherently more susceptible to the structural variations and diverse interfaces encountered in cross-app environments.

### 5.3.3. ONLINE GUI AGENT EVALUATION

**AndroidWorld.** As shown in Table 3, SE-GA exhibits strong robustness in real-world environments, achieving a success rate of 39.0%. This consistent and pronounced advantage demonstrates that SE-GA is markedly more effective at handling the tasks of dynamic environments. We attribute this improvement to the self-evolution mechanism of SE-GA, which enables the agent to continuously explore and adapt to dynamic environmental changes while leveraging past successful experiences to guide decision-making. By explicitly leveraging past successful trajectories and structured memories, SE-GA can progressively improve its decision-making policy beyond the limitations of static pretraining. In contrast, baselines such as OS-Genesis and GPT-4o primarily rely on zero-shot generalization from static pretraining. This reliance on fixed pretrained policies limits their ability to adapt to dynamic interface changes (e.g., shifts in icon layouts), resulting in reduced efficiency and lower task completion rates.

### 5.4. Ablation Study

This section evaluates the effectiveness of two core components: (1) TTME, which provides hierarchical memory retrieval during inference, and (2) MASE, which adopts a two-stage training paradigm to refine the policy. The abla-

tion results are summarized in Table 4.

**TTME enables robust long-horizon reasoning.** The TTME module is critical for completing complex multi-step tasks where maintaining long-range context is essential. While removing TTME leads to only a modest performance drop of 5.6% on short-horizon tasks (AndroidControl-Low), the impact becomes much more pronounced in long-horizon settings. Specifically, on AndroidControl-High, disabling TTME reduces the success rate from 73.8% to 61.4%, a substantial decrease of 12.4%. This result highlights that, under partial observability in extended interaction sequences, SE-GA critically depends on TTME to preserve task-relevant context and prevent catastrophic forgetting, thereby enabling coherent reasoning and planning over long episodes.

**MASE establishes the foundational capabilities for decision-making.** Removing the MASE module leads to the most severe performance degradation across all benchmarks. Compared with the full SE-GA model, the variant without MASE reduces success rates from 73.8% to 59.7% on AndroidControl-High and from 83.9% to 60.4% on GUIOdyssey. These results indicate that the memory-augmented self-evolution mechanism is essential for enabling the VLM to effectively learn from experiences stored in the memory repository, thereby substantially improving its decision-making ability when executing user instructions.

In addition, Appendix C.2 provides representative case studies, Appendix C.3 further analyzes the contributions of different modules in short-horizon and long-horizon task, and Appendix C.4 examines the roles of different components. Overall, TTME primarily boosts the success rate of SE-GA on long-horizon tasks, while MASE effectively strengthens its fundamental grounding and planning capabilities.

## 6. Conclusion

This study presents SE-GA, a unified framework designed to overcome the limitations of existing GUI agents. We first introduce the TTME module, which maintains a hierarchical memory repository consisting of episodic, semantic, and experiential memories, enabling the agent to retrieve task-relevant information for long-horizon planning in multi-step interactions. Furthermore, we propose the MASE training framework, which leverages the Hindsight Goal-Shifting

mechanism for efficient data synthesis and a GRPO-based optimization algorithm for stable continual learning. By incorporating token-level policy aggregation, hierarchical reward design, and adaptive clipping strategies, SE-GA can effectively adapt to diverse environments.

Extensive experiments across multiple benchmarks demonstrate that SE-GA achieves superior success rates and robust generalization in both offline and online GUI navigation tasks, highlighting the potential of the memory-augmented self-evolution method for building more capable and reliable GUI automation systems.

## Acknowledgements

This work was supported by the National Natural Science Foundation of China (62572346 and 62406188), and the Shanghai Municipal Special Program for Basic Research on General AI Foundation Models (2025SHZDZX025G08).

## Impact Statement

This paper presents a general framework for building more capable and robust GUI agents through self-evolution and memory-augmented decision-making. The primary goal of this work is to advance the field of machine learning and autonomous agents by improving their ability to perform long-horizon tasks in complex and dynamic environments. The techniques proposed in this paper are intended to enhance the reliability and generalization of automated systems for interacting with software interfaces, which may have positive impacts on productivity, accessibility, and the automation of repetitive digital tasks. At the same time, as with most advances in agent and automation technologies, these methods could potentially be applied in contexts that require careful consideration, such as large-scale automation or misuse in unintended scenarios. We do not foresee any immediate negative societal impacts that are specific to the methods proposed in this work beyond those generally associated with more capable automated systems. We believe that the benefits of improved robustness and adaptability in GUI agents outweigh the potential risks, and we hope this work will encourage further research on building reliable, controllable, and beneficial autonomous systems.

Despite the promising performance demonstrated by SE-GA, this work acknowledges a primary methodological limitation. Memory Retrieval Efficiency presents a potential bottleneck. As the TTME module accumulates interaction data, the scale of the hierarchical memory repository, particularly the experiential memory, grows continuously. The retrieval operations relying on embedding similarities and visual features may introduce significant computational overhead during inference, potentially hindering real-time responsiveness in latency-sensitive environments.

To address these limitations and further advance GUI agents, we identify three key directions for future research. First, we plan to scale up the training dataset to include diverse task types. Expanding beyond the current 4K trajectories to a larger corpus of interaction data will be essential to test the robustness of SE-GA and further enhance its generalization capabilities across broader scenarios. Second, we aim to explore hierarchical task decomposition for long-horizon planning. While TTME aids in context management, integrating explicit sub-goal decomposition strategies could significantly improve the agent's ability to reason through and execute ultra-long workflows that span multiple applications. Finally, we intend to investigate transfer learning across different GUI platforms. Future work will assess how effectively the evolved policies and memory structures adapt to distinct platform nuances—spanning mobile, web, and desktop interfaces—thereby moving closer to the goal of building truly universal GUI agents.

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

# A. Dataset Details

To ensure comprehensive evaluation, this study utilizes four diverse benchmarks covering static grounding, offline instruction execution, and online dynamic interaction. Detailed statistics and configurations for each dataset are provided below.

## A.1. ScreenSpot

ScreenSpot is a comprehensive benchmark designed to evaluate the GUI grounding capabilities of Large Multimodal Models in translating text-based instructions into precise visual locations. It features over 1,200 instructions spanning five major operating environments: iOS, Android, macOS, Windows, and Web, offering a diverse and realistic testbed for cross-platform generalization. Unlike datasets relying on synthetic generation or view hierarchies, ScreenSpot comprises high-quality screenshots curated by human researchers based on typical daily usage scenarios, ensuring high relevance and visual complexity. A key strength lies in its fine-grained element classification, distinguishing between Text and Icon/Widget types to rigorously evaluate distinct visual recognition skills. The dataset provides ground truth in the form of bounding boxes $(x_{min}, y_{min}, x_{max}, y_{max})$, enabling precise zero-shot evaluation of a model's ability to locally ground elements across varying screen resolutions and layouts. Following the settings from previous work (Wang et al., 2025a), this study evaluates the grounding accuracy of GUI agents on both textual and icon-based elements across these diverse environments.

## A.2. AndroidControl

AndroidControl is a large-scale dataset designed to rigorously measure the ability of agents to generalize beyond the apps and tasks they were trained on. It features over 15,000 demonstrations collected from human raters, covering 833 distinct applications across 40 diverse categories on Android devices. Unlike datasets limited to specific domains or synthetic environments, AndroidControl offers high-quality execution traces including high-fidelity screenshots, accessibility trees, and dual-granularity natural language instructions (high-level goals and low-level steps). A key strength lies in its structural design, which specifically targets the evaluation of out-of-domain generalization in dynamic mobile environments. The dataset's comprehensive action space includes eight core operations: CLICK, LONG_PRESS, SCROLL, OPEN_APP, INPUT_TEXT, NAVIGATE_HOME, NAVIGATE_BACK, and WAIT, capturing the full spectrum of interactions required for autonomous mobile control. Following established settings (Li et al., 2024), this study assesses the model on out-of-domain data in low-level instruction execution and high-level planning scenarios, reporting action type accuracy, grounding accuracy, and success rate.

## A.3. GUIOdyssey

GUIOdyssey is a comprehensive dataset designed to train and evaluate cross-app navigation agents capable of executing complex workflows across multiple applications. It features 7,735 episodes spanning 201 apps and over 1,400 application combinations across 6 distinct mobile devices, offering a diverse and high-fidelity environment for training. Unlike single-app or single-device datasets, GUIOdyssey incorporates varied hardware profiles—including Pixel Fold, Tablet, and standard phones—providing rich data like device-specific screenshots and metadata. A key strength lies in its four distinct test splits—random, task, device, and app—designed to rigorously evaluate the agent's generalization capabilities across unseen applications, tasks, and hardware form factors. The dataset's action space includes eight core operations: CLICK, SCROLL, LONG_PRESS, TYPE, COMPLETE, IMPOSSIBLE, HOME, and BACK, capturing the full spectrum of interactions required for dynamic cross-app navigation. Following established settings (Xu et al., 2025), this study randomly samples 500 episodes to create a consistent evaluation subset and reports the action type accuracy, grounding accuracy, and step success rate.

## A.4. AndroidWorld

AndroidWorld is a dynamic environment designed for building and benchmarking autonomous computer control agents on a live Android emulator. It features a highly reproducible benchmark of 116 hand-crafted tasks across 20 real-world applications, utilizing dynamic instantiation with randomly-generated parameters to create millions of unique task variations. Unlike static datasets, AndroidWorld interacts with a live operating system, offering an open environment with access to millions of apps and websites while maintaining a lightweight computational footprint. A key strength lies in its reliable evaluation framework, which employs durable reward signals to ensure consistent benchmarking scores even in a live environment. The platform is designed for extensibility, and the easy addition of new tasks to rigorously evaluate agent adaptability.

### A.5. The detail of used baseline

During our tests, we referred to the test data from the UI-TARS paper including UI-TARS, GPT-4o, Gemini, Claude3, InternVL, Aria-UI, Aguvis, UGround, and SeeClick. The rationality of using external results is as follows: 1. Reproducibility: The training data and infrastructure used in some benchmark tests cannot be fully replicated. Using the values they have published ensures a fair comparison with their official benchmarks. 2. Benchmark consistency: All the evaluation results in this paper are obtained on the same benchmark version following the same evaluation process, thus allowing for direct comparison and being reasonable.

### A.6. Detailed information of the training dataset

*Table 5.* The detailed information of the dataset

| Metric | Value |
| --- | --- |
| Total trajectories | 4,007 |
| Average steps per trajectory | 11.89 |
| Minimum trajectory length | 1 |
| Maximum trajectory length | 31 |
| Median trajectory length | 12 |
| Short Task (1-7 steps) | 67% |
| Medium Task (8-15 steps) | 22% |
| Long Task (16-47 steps) | 11% |

## B. Training Details

### B.1. Implementation Framework

The training pipeline of this study used in MASE framework is built upon the DeepSpeed library to maximize computational efficiency and memory optimization.

- **DeepSpeed Configuration:** This study utilizes ZeRO Stage 3 (Zero Redundancy Optimizer) to partition optimizer states, gradients, and parameters across GPUs. To ensure training stability and speed, this study disables CPU offloading (`"offload_optimizer": "device": "none"`) and enable BF16 mixed precision training.

- **Parameter Efficient Fine-Tuning (PEFT):** For the Self-Evolution Training (Stage II), this study employs Low-Rank Adaptation (LoRA) to efficiently update the policy model while freezing the main backbone. This approach significantly reduces the GPU memory footprint during the reinforcement learning phase.

### B.2. Hyperparameter Configuration

Table 6 details the specific hyperparameters used for the Self-Evolution (RL) stage. This study sets the maximum context window to handle long-horizon GUI trajectories, with a prompt length of 6144 tokens to accommodate hierarchical memory contexts (episodic, semantic, and experiential) and high-resolution screen observations.

*Table 6.* Detailed Hyperparameters for Stage II: Self-Evolution Training (RL).

| Category | Configuration / Value |
|---|---|
| *LoRA Configuration* | |
| Task Type | CAUSAL_LM |
| Rank ($r$) | 32 |
| Alpha ($\alpha$) | 64 |
| Target Modules | all-linear |
| Dropout | 0.05 |
| Bias | none |
| *DeepSpeed (ZeRO-3) Settings* | |
| Stage | 3 |
| Precision | BF16 (enabled: auto) |
| Offload Optimizer | None (GPU Only) |
| Offload Param | None (GPU Only) |
| Overlap Comm | True |
| Contiguous Gradients | True |
| Sub-group Size | $1 \times 10^9$ |
| *Context & Sequence* | |
| Max Prompt Length | 6144 |
| Max Completion Length | 1024 |

## B.3. Prompt Templates

To facilitate the structured reasoning required for the Test-Time Memory Extension (TTME), this study utilizes a system prompt that explicitly instructs the model to retrieve and utilize memory before taking action. Some key parts of the prompt template used during inference are shown below:

System Prompt for SE-GA

You are an AI assistant designed to simulate the model reasoning process for UI navigation tasks. Your goal is to generate a rigorous reasoning chain before a specific action is executed.

Based on the `Task Instruction`, `Current Screenshot` context, `Previous History Summary`, `Upcoming Action`, and `Thought`, you must strictly adhere to the following reasoning steps:

1. **Progress Assessment**: Analyze the current interface state and evaluate the progress toward the goal.

2. **Decision Rationale**: Formulate the strategy and justify the upcoming action.

3. **History Summary**: Update the history summary to include the execution of the upcoming action.

### Output Format:

```
<Progress_Evaluation>
...  (One or two sentences)
</Progress_Evaluation>
<Decision_Rationale>
...  (One or two sentences)
</Decision_Rationale>
<History_Summary>
...  (One or two sentences)
</History_Summary>
<Answer
{{'action': '', 'value': '', 'position': []}}
<Answer>
```

### Example:
**Input:** Task Instruction: Find all events in New York City in September.
Previous History Summary: The user first set the location to New York, then set the start date to September 1st and the end date to September 30th.
**Output:**

```
<Progress_Evaluation>
The user has successfully set the location to New York and the date range
to Sept 1-30, but the displayed events are still from March, indicating the
date filter needs to be applied.
</Progress_Evaluation>
<Decision_Rationale>
Clicking the "Apply" button will confirm the selected date range (Sept 1-30)
and refresh the event list to show only activities occurring in New York
City during September.
<Decision_Rationale>
<History_Summary>
The user changed the location to New York, set the date range to September
1st-30th, and applied the filter to update the event list.
<History_Summary>
<Answer
{{'action': 'click', 'value': 'Apply', 'position': [0.3, 0.66]}}
<Answer>
```

### Current Input:
Task Instruction: {_TASK}
Previous History Summary: {_MEMO}
### Current Output:
Current Action: {_ACTION}
Thought: {_THOUGHT}

# C. Case Study

## C.1. Example of Hindsight Goal-Shifting

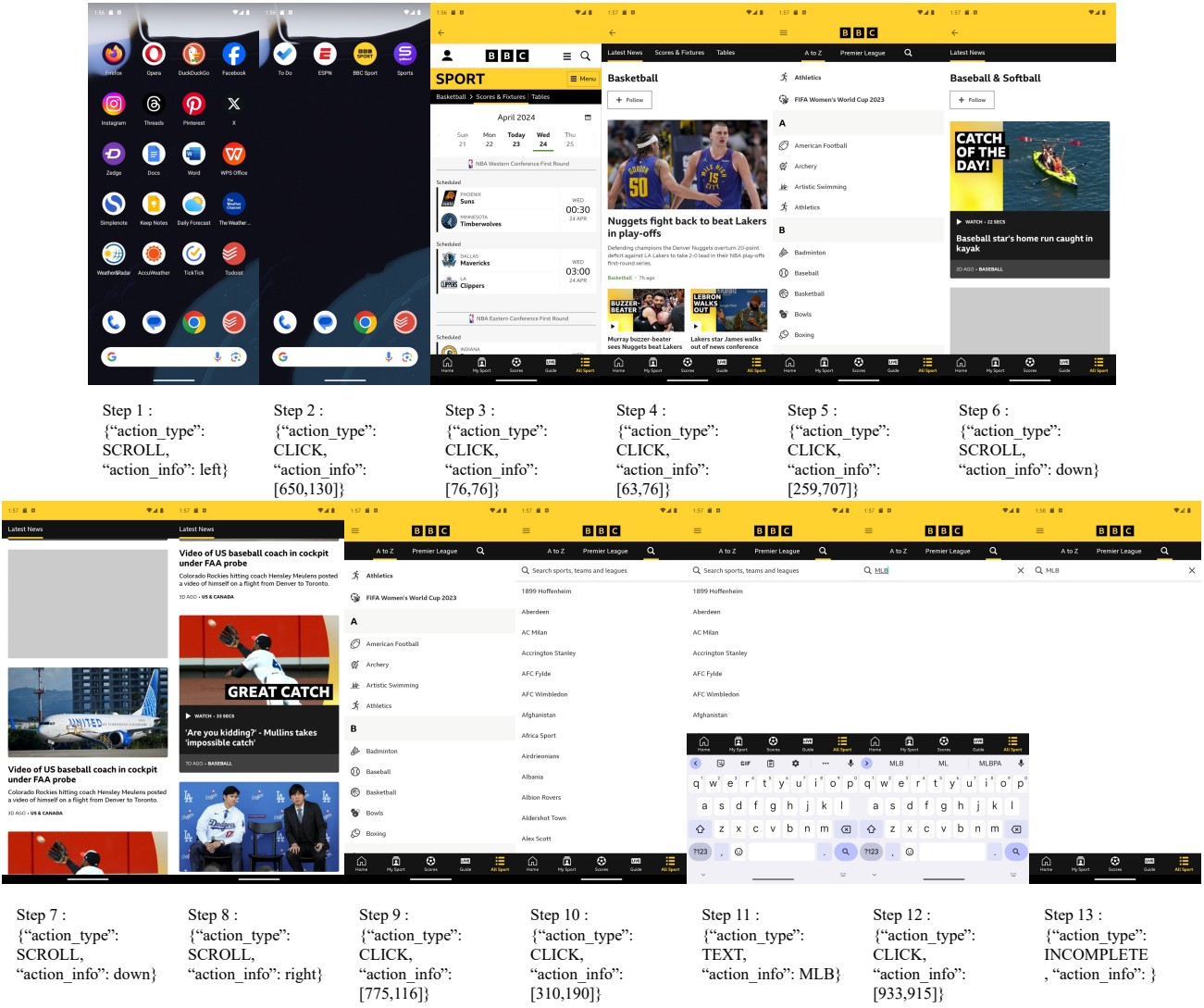

*Figure 2.* A failure trajectory example. The task instruction is "Using BBC Sports, find out when the next MLB game is scheduled and then create a reminder in Microsoft To Do."

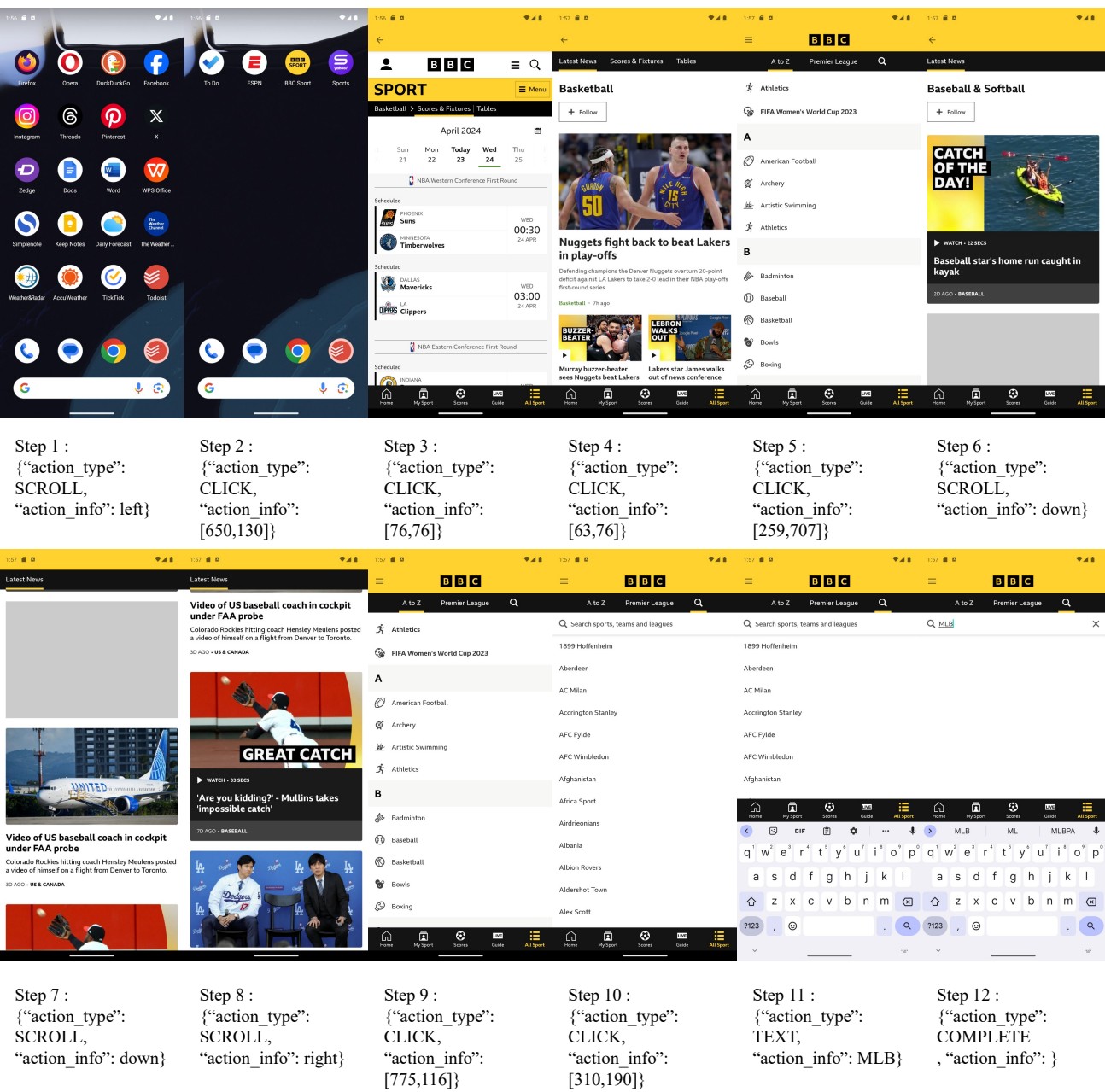

*Figure 3.* A successful trajectory example. By using Hindsight Goal-Shifting, the agent successfully discovers and executes a sequence of actions that complete the assigned task. The new task instruction is "Using BBC Sports, find out the next MLB game in the search bar."

## C.2. Long-horizon Task Case

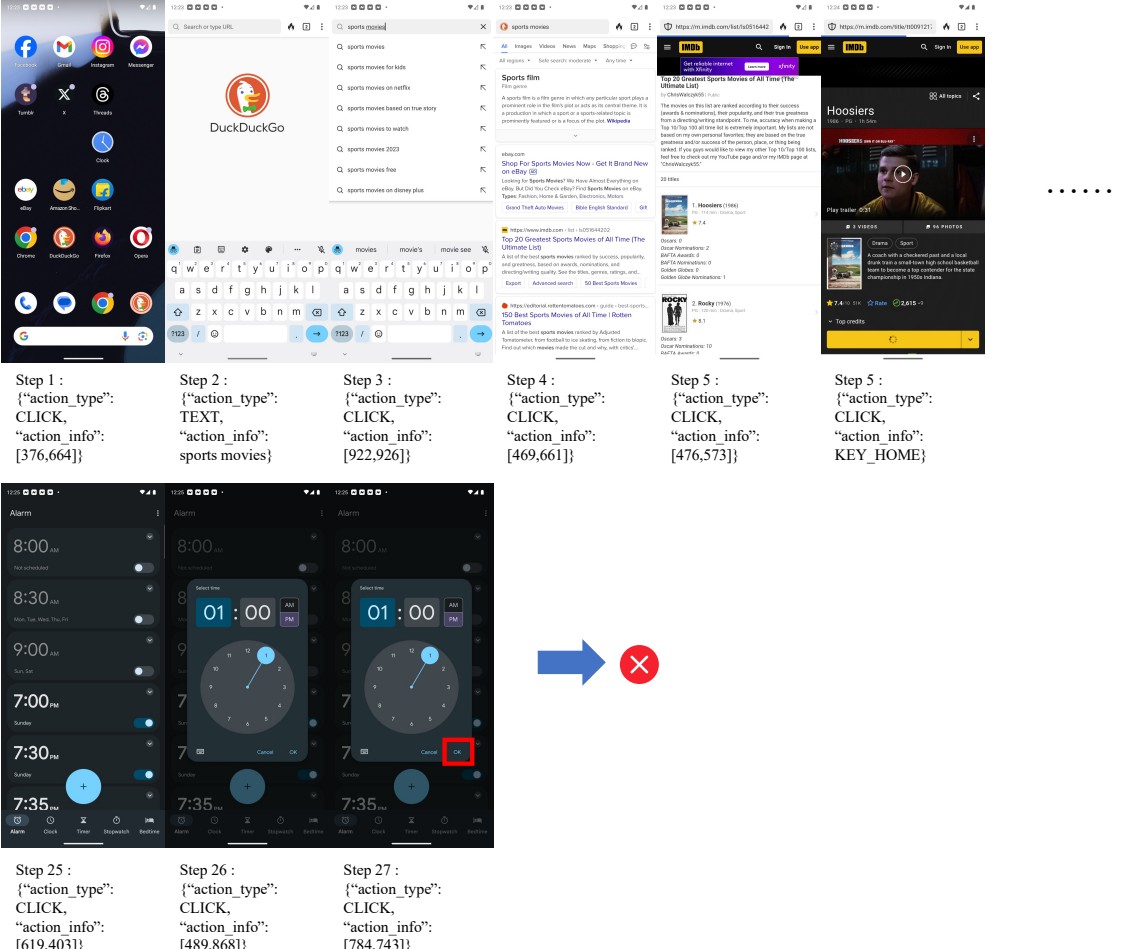

Step 1 :
{"action_type":
CLICK,
"action_info":
[376,664]}

Step 2 :
{"action_type":
TEXT,
"action_info":
sports movies}

Step 3 :
{"action_type":
CLICK,
"action_info":
[922,926]}

Step 4 :
{"action_type":
CLICK,
"action_info":
[469,661]}

Step 5 :
{"action_type":
CLICK,
"action_info":
[476,573]}

Step 5 :
{"action_type":
CLICK,
"action_info":
KEY_HOME}

Step 25 :
{"action_type":
CLICK,
"action_info":
[619,403]}

Step 26 :
{"action_type":
CLICK,
"action_info":
[489,868]}

Step 27 :
{"action_type":
CLICK,
"action_info":
[784,743]}

*Figure 4.* A failure trajectory example of UI-TARS. The task instruction is "Plan an evening of sports-themed entertainment by selecting a sports movie using DuckDuckgo and adding some snacks to your Amazon shopping cart. Invite Victor James through Facebook Messenger, and set a reminder on your Clock app so you don't forget."

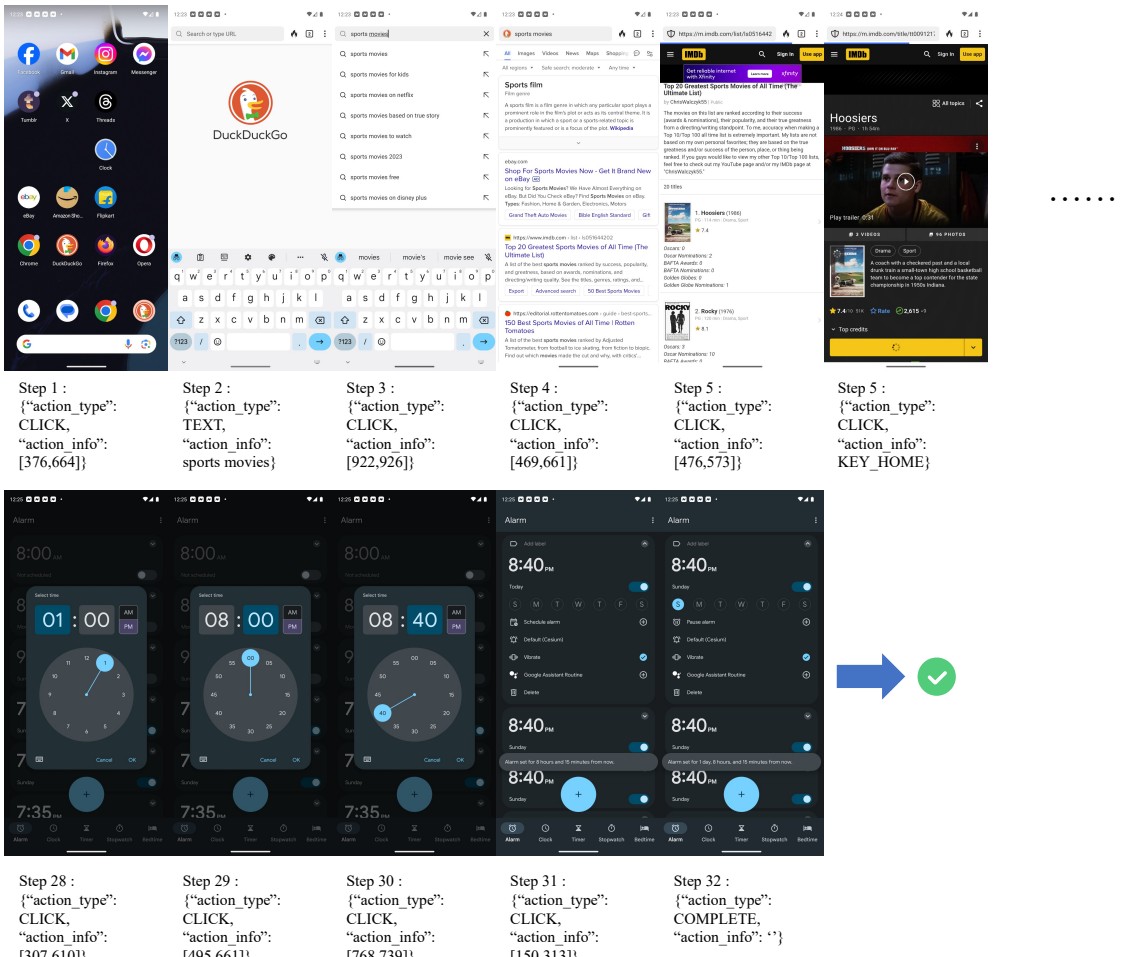

*Figure 5.* A successful trajectory example of SE-GA. The task instruction is "Plan an evening of sports-themed entertainment by selecting a sports movie using DuckDuckgo and adding some snacks to your Amazon shopping cart. Invite Victor James through Facebook Messenger, and set a reminder on your Clock app so you don't forget."

## C.3. Ablation Study about Short-horizon Task and Long-horizon Task

To further dissect the robustness of SE-GA, Fig. 6 visualizes the success rate trajectories across tasks with varying horizon lengths ($T = 0 \rightarrow 30+$ steps) on benchmark GUIOdyssey. Notably, while baseline variants exhibit varying degrees of performance decay as complexity increases, SE-GA demonstrates exceptional stability, maintaining a high success rate even over ultra-long trajectories. This comparison underscores the distinct mechanisms of our proposed modules:

- **TTME for Long-Horizon Consistency:** The exclusion of the Test-Time Memory Extension (w/o TTME) leads to a pronounced performance deficit that widens as the task length extends. This confirms that partial observability is the primary bottleneck in multi-step reasoning. In the absence of hierarchical memory to actively retrieve and preserve critical historical context, the agent suffers from attention dilution, losing track of early sub-goals in later steps. TTME effectively bridges this gap, ensuring that long-term dependencies are accurately maintained.

- **MASE for Foundational Robustness:** The removal of the Memory-Augmented Self-Evolution (w/o MASE) undermines the fundamental decision-making capabilities of the agent. Without the policy refinement and the Hindsight Goal-Shifting mechanism, the agent acts as a static executor lacking the adaptability to recover from local errors. This deficiency limits its fundamental performance across the board, validating that self-evolution is essential for GUI agents in dynamic GUI environments.

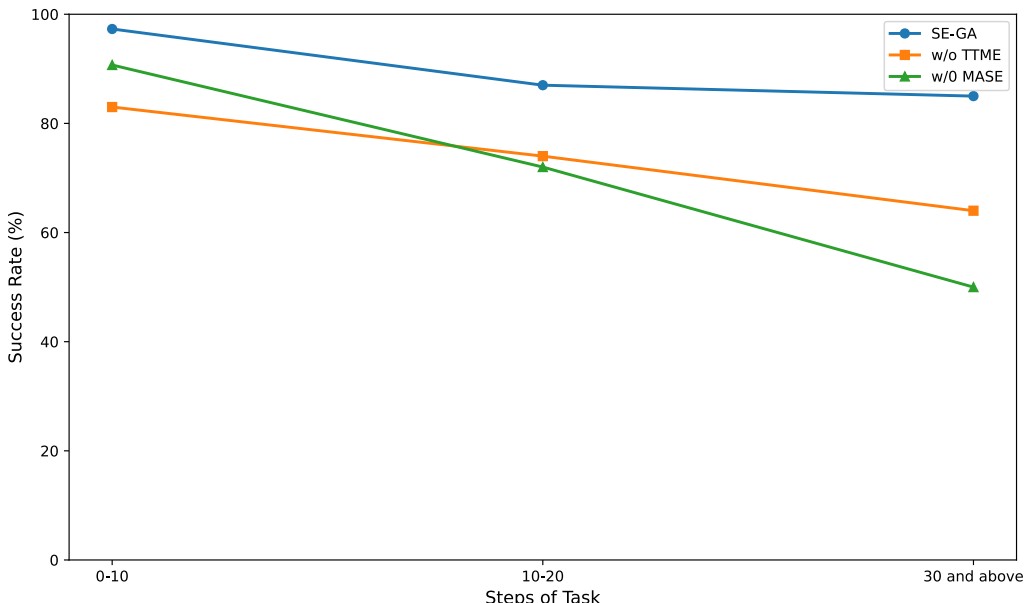

*Figure 6.* SE-GA Performance on Different Task Steps.

## C.4. Detailed ablation experiments

*Table 7.* Extended Ablation Study

| Method | AndroidControl-High | | | GUIOdyssey | | |
|---|---|---|---|---|---|---|
| | Type | Grounding | SR | Type | Grounding | SR |
| SE-GA | 94.6 | 92.5 | 88.6 | 84.4 | 76.4 | 73.8 |
| w/o TTME | 91.9 | 90.8 | 83.0 | 81.0 | 68.4 | 61.4 |
| w/o MASE-Stage II | 88.6 | 86.9 | 74.3 | 72.2 | 60.1 | 59.7 |
| w/o MASE-Stage I | 79.0 | 77.2 | 70.6 | 68.7 | 59.3 | 44.0 |
| w/o MASE | 69.7 | 59.1 | 55.7 | 66.4 | 51.8 | 49.3 |

## C.5. Performance Across Multiple Rounds of Self-Evolution

To evaluate the continual self-improvement capability of the proposed framework, we conduct experiments over three consecutive rounds of self-evolution. In each round of the experiment, the agent first interacted with the environment through the TTME module to collect new interaction trajectories, then constructs the self-evolution dataset $\mathcal{D}_{\text{EVO}}$ using the Hindsight Goal-Shifting mechanism, and finally updates its policy via the MASE training pipeline.

*Table 8.* Performance across multiple rounds of self-evolution. The results show consistent improvements on all benchmarks, demonstrating the continual self-evolution capability of SE-GA.

| Benchmark | Round 1 | Round 2 | Round 3 |
|---|---|---|---|
| ScreenSpot | 79.3 | 86.0 | 89.0 |
| AndroidControl-Low | 68.3 | 75.5 | 88.6 |
| AndroidControl-High | 55.9 | 71.3 | 75.8 |
| GUIOdyssey | 52.3 | 75.1 | 83.9 |
| AndroidWorld | 28.6 | 34.5 | 39.0 |

As shown in Table 8, SE-GA consistently improves across all evaluated benchmarks from Round 1 to Round 3, demonstrating its capability for continual self-evolution. Several key observations can be drawn from these results.

**Substantial improvements on long-horizon and complex tasks.** The most significant gains are observed on benchmarks requiring long-horizon planning and cross-application interaction. For example, on GUIOdyssey, the success rate increases from 52.3% in Round 1 to 75.1% in Round 2, and further improves to 83.9% in Round 3. Similarly, AndroidControl-High achieves a notable gain of 15.4% after the first evolution round. We attribute these improvements to the continual accumulation of high-quality experiential memory and the Hindsight Goal-Shifting mechanism within MASE. As the agent successfully completes more intermediate sub-goals, the memory repository gradually stores increasingly diverse and reliable trajectories, providing effective guidance for handling extended interaction sequences and reducing error propagation.

**Progressive refinement of grounding and low-level execution capabilities.** SE-GA also exhibits stable improvements on ScreenSpot and AndroidControl-Low, with performance increasing from 79.3% to 89.0% and from 68.3% to 88.6%, respectively, over the three evolution rounds. These results suggest that MASE, particularly the Hierarchical Reward Design, progressively strengthens the agent's visual grounding and low-level action execution abilities through iterative self-training, rather than merely reinforcing high-level behavioral patterns.

**Effective generalization to dynamic environments.** Performance on the online AndroidWorld benchmark also improves steadily. Although the absolute gains are smaller than those observed on offline benchmarks due to the higher volatility and partial observability of real-world environments, the consistent upward trend indicates that SE-GA can effectively transfer previously acquired successful experiences to unseen dynamic states, thereby continuously adapting its policy beyond static pretraining.

Furthermore, the rate of improvement gradually decreases from Round 2 to Round 3. This phenomenon is likely caused by policy convergence and the diminishing marginal utility of newly collected trajectories as the memory repository becomes increasingly saturated with similar successful experiences. Overall, these results demonstrate that SE-GA can progressively evolve from a static task executor into a continually improving autonomous agent.

## C.6. Additional Experiments

We further conduct ablation experiments to investigate the impact of different retrieval strategies in the memory module. The results are summarized in Table 9.

*Table 9.* Comparison of different retrieval strategies

| Strategy | AndroidControl-High | GUIOdyssey |
|---|---|---|
| Top-k | 75.8 | 83.9 |
| Mixed | 76.2 | 82.1 |
| Success-only | 70.0 | 72.5 |

The results show that both the Top-k and Mixed retrieval strategies consistently outperform the Success-only strategy across

all benchmarks, indicating that failed trajectories also provide valuable supervisory signals for decision-making and error correction.

Notably, the performance gap between the Top-k and Mixed strategies is relatively small. We attribute this to the text-image hybrid retrieval mechanism proposed in TTME, which can naturally retrieve a balanced set of both successful and failed trajectories, thereby providing sufficient diversity for effective reasoning without the need for explicit sampling ratio limitations.

### C.7. Some Specific Examples of Using Memory

To further illustrate how different memory types in TTME support inference and decision-making, we provide several representative examples demonstrating how episodic memory, semantic memory, and experiential memory are retrieved and utilized during GUI interaction.

**Example 1: Episodic Memory ($M^{EPI}$) for Short-Term Sequential Reasoning.**

*Task:* "Open the Settings application and enable battery saver mode."

Suppose the agent has already executed the following interaction trajectory during the current task:

$$m_1 : \langle \texttt{HomeScreen, click(Settings), SettingsPage} \rangle,$$
$$m_2 : \langle \texttt{SettingsPage, scroll(Down), BatterySection} \rangle.$$

At the current step $t$, the episodic memory repository stores these recent transitions within the sliding window horizon $H$:

$$\mathcal{C}_t^{epi} = [m_1, m_2].$$

During reasoning, the agent retrieves $\mathcal{C}_t^{epi}$ to infer the current navigation progress. Since the recent history indicates that the agent has already entered the battery-related settings page, the policy avoids redundant navigation actions such as returning to the home page or reopening Settings. Instead, the agent directly predicts the next relevant action:

$$a_t = \texttt{click(BatterySaverToggle)}.$$

This example demonstrates that episodic memory primarily functions as a short-term working memory that preserves recent interaction context, enabling coherent multi-step decision-making and preventing repetitive or contradictory actions.

**Example 2: Semantic Memory ($M^{SEM}$) for General Interaction Rules.**

*Task:* "Access the personal order history page in a shopping application."

Assume that the semantic memory repository contains the following abstract interaction rule:

"Users typically need to log in before accessing personal account pages or order history."

Formally, this semantic entry is represented as:

$$m_i^{sem} = \langle k_i^{sem}, d_i \rangle,$$

where $d_i$ corresponds to the above rule description.

Given the current instruction $Q$, the retrieval module computes the similarity score:

$$S^{sem}(Q, m_i^{sem}),$$

During reasoning, the retrieved semantic context $\mathcal{C}^{sem}$ guides the agent to first verify whether the current application state has already been authenticated. If the user is not logged in, the agent will take the following actions first instead of directly searching for the order history:

$$\text{click(Login)} \rightarrow \text{text(Account)} \rightarrow \text{text(Password)}.$$

This example shows that semantic memory provides persistent task-general knowledge that helps the agent understand high-level interaction logic beyond the current trajectory.

**Example 3: Experiential Memory ($M^{EXP}$) for Reusing Historical Task Strategies.**

*Task:* "Download a PDF attachment from Gmail and upload it to a reimbursement application."

Suppose the experiential memory repository contains a previously successful trajectory:

$$\tau_i = \{\text{Open Gmail} \rightarrow \text{Download Attachment} \rightarrow \text{Open File Manager} \rightarrow \text{Upload PDF}\}.$$

The corresponding reflective summary generated by the agent is:

"For reimbursement tasks, PDF files are usually stored in the Downloads folder after attachment extraction. Uploading directly from Downloads is more reliable than selecting recent files."

During inference, the retrieval system jointly considers both the semantic intent of the instruction and the visual similarity of the current GUI observation:

$$\begin{aligned} S^{exp}(Q, o_t) =& \lambda \cdot \text{Sim}(\phi(Q), k_i^{intent}) \\ &+ (1 - \lambda) \cdot \text{Sim}(\psi(o_t), k_i^{task}). \end{aligned}$$

Since both the task objective and the current Gmail interface are highly similar to the stored experience, this trajectory is retrieved into $\mathcal{C}^{exp}$.

The retrieved reflective summary then influences reasoning by encouraging the agent to navigate directly to the Downloads folder during the upload stage, instead of repeatedly searching across unrelated directories. In this way, experiential memory enables the agent to reuse previously successful execution strategies for handling similar long-horizon tasks.

**Example 4: Collaborative Usage of Multiple Memory Types.**

*Task:* "Book a flight ticket and save the itinerary screenshot."

During inference, all three memory systems collaborate simultaneously:

- Episodic memory ($M^{EPI}$) tracks the recent navigation history, ensuring that the agent remembers whether it has already selected departure dates or passenger information.

- Semantic memory ($M^{SEM}$) provides general rules such as:

    "Flight booking usually requires selecting departure city, destination, date, and passenger information before payment."

- Experiential memory ($M^{EXP}$) retrieves successful historical booking trajectories and reflective summaries, such as:

    "The itinerary screenshot is typically displayed after payment confirmation and can be captured before closing the booking page."

By jointly leveraging short-term context, abstract interaction knowledge, and historical task experiences, the agent performs more reliable long-horizon reasoning and avoids common execution failures.

Overall, the three memory types in TTME serve complementary roles during inference. Episodic memory maintains recent interaction continuity, semantic memory provides transferable interaction knowledge, and experiential memory enables the reuse of successful historical strategies. Their coordinated retrieval and integration collectively improve the agent's reasoning capability and robustness in complex GUI environments.

