# OpenReview forum: "SE-GA: Memory-Augmented Self-Evolution for GUI Agents"
_ICML.cc/2026/Conference — ICML 2026 regular_

### Official Review · Reviewer_rRHX · 2026-03-02

**Soundness:** 3
**Presentation:** 3
**Significance:** 4
**Originality:** 2
**Overall Recommendation:** 3
**Confidence:** 5

**Summary:**

This paper proposes SE-GA, a memory-augmented self-evolution framework for GUI agents integrating hierarchical memory structures with iterative self-improvement. The authors introduce TTME for test-time memory retrieval and MASE, a two-stage training pipeline with hindsight goal-shifting.

**Compliance With Llm Reviewing Policy:**

Affirmed.

**Final Justification:**

The rebuttal improves the clarity of the paper and partially addresses my concerns, but the core issues are still not fully resolved. In particular, the proposed retraining trigger is now better supported by additional analysis, yet it still appears to be a system-specific heuristic rather than a sufficiently justified general principle. Moreover, the experimental pipeline remains insufficiently transparent, especially with respect to the multi-round evolution setup, the composition and updating of the training data, the verifier mechanism used to determine success, and how memory is handled across rounds.

**Key Questions For Authors:**

Related Work Clarification:

(1) Can the authors further clarify the differences between this work and GUI-Explorer: Autonomous Exploration and Mining of Transition-aware Knowledge for GUI Agent, as well as OS-Genesis: Automating GUI Agent Trajectory Construction via Reverse Task Synthesis?

(2) The hierarchical memory mechanism in TTME appears similar to that in Agent S: An Open Agentic Framework that Uses Computers Like a Human. Can the authors clearly articulate the technical distinctions?

Experimental Setup:

(1) The memory repository is pre-populated with existing datasets rather than built from scratch through autonomous exploration. Can the authors provide experiments demonstrating true autonomous evolution in a limited environment, similar to MemGUI-Bench: Benchmarking Memory of Mobile GUI Agents in Dynamic Environments?

(2) The MASE pipeline involves periodic model retraining. What is the practical criterion for determining when to trigger training in long-term deployment?


References:

(1) GUI-Explorer: Autonomous Exploration and Mining of Transition-aware Knowledge for GUI Agent

(2) OS-Genesis: Automating GUI Agent Trajectory Construction via Reverse Task Synthesis

(3) Agent S: An Open Agentic Framework that Uses Computers Like a Human

(4) MemGUI-Bench: Benchmarking Memory of Mobile GUI Agents in Dynamic Environments

**Limitations:**

See Key Questions For Authors.

**Strengths And Weaknesses:**

1. The hierarchical memory design provides a principled approach to long-horizon context management in GUI navigation.

2. Strong empirical results across multiple benchmarks validate the effectiveness, particularly on long-horizon tasks.

---

> ### Author Rebuttal · Authors · 2026-03-30
>
> We sincerely thank the reviewer for the detailed and constructive feedback. We are encouraged by your recognition of our framework’s potential for addressing long-term tasks in GUI agents. We believe our response has addressed most of your concerns. Below, we provide detailed responses to all the comments:
>
> ---
> **Response to Q(1):**
> A1: Thank you for this insightful question. We appreciate the opportunity to clarify the technical distinctions between SE-GA and these recent works.
> - **vs. GUI-Explorer:** GUI-Explorer focuses on exploring and storing transition knowledge in a **static knowledge** base. In contrast, SE-GA aims for **policy evolution** through TTME and MASE framework. Instead of merely retrieving external knowledge, SE-GA explicitly encodes non-parametric experiences into parametric policy weights, which can continuously enhance the model's own capabilities.
> - **vs. OS-Genesis:** OS-Genesis generates training data via offline **Reverse Task Synthesis**. SE-GA adopts online **Hindsight Goal-Shifting**, relabeling failed trajectories from live execution to handle **dynamic failures**. SE-GA significantly outperforms OS-Genesis on the dynamic **AndroidWorld** benchmark (39.0% vs. 17.4%), which validates the effectiveness of our framework over static paradigms.
> ---
> **Response to Q(2):**
> A2: Thank you for this valuable comparison. While both utilize hierarchical structures, they differ fundamentally in both **design philosophy** and **evolutionary capability**.
> - **Design philosophy:** Agent S organizes memory for task decomposition. SE-GA’s TTME is cognitively-inspired, categorizing memory into **Episodic**, **Semantic**, and **Experiential**. Notably, our **Semantic Memory** encodes universal interaction logic (e.g., "Log in before access"), which is a capability that Agent S lacks.
> - **Evolutionary capability:** Agent S uses memory for inference-time retrieval. However, the memory curated within SE-GA is not just retrieved but used to update the model’s policy parameters, bridging the gap between non-parametric memory and parametric policies to enable **self-evolution** in model strategies.
> ---
> **Response to Q(3):**
> A3: Thank you for this insightful comment. We apologize for the ambiguity regarding our experimental setup in the original submission.
> - **Clarification on Memory Construction:** During the **experimental evaluation**, the memory repository was **built from scratch through autonomous exploration**. It was **not pre-populated**. The "existing datasets" mentioned in Section 5.1 were used solely to construct the **training data for the MASE policy optimization (Stage I & II)**, and not to initialize the memory buffer during testing. We will explicitly state this distinction in the final version.
> - **Regarding MemGUI-Bench:** We fully acknowledge the value of MemGUI-Bench for evaluating GUI agents. However, as MemGUI-Bench was not publicly available prior to our submission, we were unable to include it in our initial experiments. Here, we have provided some additional experiments for your reference.
>
> >R4-Table 1: Performance on MemGUI-bench
>
> |Agent|Success Rate|
> |---|---|
> |SE-GA|12.5|
> |UI-TARS|8.3|
> |CogAgent|0.0|
> - **Evidence of Autonomous Evolution:** We believe our current results on **AndroidWorld**, which share similarities with the scope of MemGUI-Bench, have demonstrated "true autonomous evolution" in a dynamic, limited environment to a certain extent. The agent's significant performance improvement (39.0% success rate) validates that it can effectively build memory and adapt autonomously from scratch.
> ---
> **Response to Q(4):**
> A4: Thank you for this practical question regarding the deployment logistics of our framework. In long-term deployment, the MASE training pipeline is triggered when the memory repository reaches a certain data volume. In practice, this threshold is a hyperparameter determined by engineering resources and can be selected based on the specific scenario. Specifically, since we conducted experiments using 4 A800 GPUs, we set the trigger threshold $\tau_{trigger}$ to 2000 to balance stability, diversity, and efficiency:
> - **Training Stability:** The algorithm used in Stage II of MASE suffers from high variance. A buffer of 2,000 trajectories minimizes the gradient variance $\text{Var}(\nabla \mathcal{L})$, ensuring stable policy convergence.
> - **Data Diversity:** This scale allows the **Hindsight Goal-Shifting** mechanism to curate a diverse dataset covering a wide state distribution $P(s)$, preventing overfitting to sparse interactions.
> - **Computational Efficiency:** Batch accumulation amortizes the training cost $C_{train}$, balancing the demand for model timeliness and resource limitations.
> ---
> We hope the new experiments and detailed justifications sufficiently address your concerns. If you find our responses convincing, we would respectfully ask you to reconsider your rating. We look forward to incorporating your valuable insights into the final version.

---

> > ### Author Rebuttal · Reviewer_rRHX · 2026-04-03
> >
> > Thank you for the detailed rebuttal. The additional clarifications are helpful, especially the note that the test-time memory is built from scratch rather than pre-populated. However, my main concerns are still only partially addressed.
> >
> > First, regarding the retraining criterion, the response explains that MASE is triggered when the memory buffer reaches 2,000 trajectories. This is useful for reproducibility, but it still appears to be an implementation heuristic rather than a sufficiently justified principle for long-term deployment. I had hoped to see either sensitivity analysis over different thresholds or a clearer discussion of when retraining should be triggered in a more principled way.
> >
> > Second, I still find the experimental design unclear. In particular, it remains hard to understand how the open-source data and the simulator-collected data are handled, where the simulator task instructions come from, how task success is determined during simulator collection, and how the multi-round evolution experiments are actually set up. It is still unclear what changes across rounds, whether new memory data are collected each round, and whether the same 4K tasks are reused or not. Since these details are central to the self-evolution claim, I believe they should be described much more explicitly.
> >
> > Overall, while the rebuttal improves the paper, my concerns about the retraining criterion, the autonomous-evolution claim, and the clarity of the experimental setup are still not fully resolved. Therefore, I am not inclined to substantially revise my overall assessment based on the current rebuttal alone.

---

> > > ### Author Response · Authors · 2026-04-04
> > >
> > > We sincerely thank the reviewer for the constructive feedback on our rebuttal. We appreciate the acknowledgement regarding the memory construction issue. To fully resolve the remaining concerns about the retraining criterion and the experimental design, we provide the following detailed clarifications.
> > >
> > > ---
> > > **1. Clarification on Retraining Criterion**
> > > We have described the data construction pipeline and the memory buffer mechanism in **Sec. 5.1** of the current manuscript. During the long-term deployment, we set the triggering conditions according to two principles:
> > > * **Memory Saturation**: As the agent interacts with dynamic environments, the TTME functions as a dynamic buffer for online evolution. However, there is a threshold for the memory repository. When this threshold is exceeded, the retrieval cost of non-parametric experience begins to exceed the value it can obtain for completing the task, and these experiences need to be refined into the internal strategy parameters $\theta$ of the model to reduce the retrieval cost. Here, we present the unit time for retrieving memories at different levels in the TTME system, as well as the total retrieval time for performing one action:
> > >     >Table: Retrieval Delay Analysis
> > >
> > >     |Buffer Size|Single retrieval time|Total Time|
> > >     |---|---|---|
> > >     |1000|~75ms|~459ms|
> > >     |1500|~146ms|~776ms|
> > >     |1750|~247ms|~1282ms|
> > >     |2000|~391ms|~1623ms|
> > >     |2250|~445ms|~2270ms|
> > >
> > >     We selected a buffer size that results in a total retrieval time ranging from 1s to 2s. If there were a better retrieval mechanism, this threshold would increase accordingly.
> > >
> > > * **Training Stability**: Besides the issue of retrieval time mentioned above, newly collected data samples often have high variance and strong temporal correlation. Retraining with an insufficient buffer size prevents the gradient distribution from being smoothed, leading to severe policy oscillation or even degradation. To clearly and intuitively demonstrate this, we conducted a **sensitivity analysis** on our devices. Specifically, we analyzed the training loss across different buffer thresholds:
> > >     >Table: Sensitivity Analysis on Retraining Buffer Size
> > >
> > >     |Buffer Size|Std|Mean|Max Spike|
> > >     |---|---|---|---|
> > >     |1500|~0.010|~0.0070|0.2856|
> > >     |1750|~0.008|~0.0060|0.2537|
> > >     |2000|~0.006|~0.0045|0.2286|
> > >     |2250|~0.005|~0.0055|0.2443|
> > >
> > >     Based on these two considerations, we selected 2000 as the threshold.
> > >
> > > The **Retrieval Delay Analysis** and **Training Stability Sensitivity Analysis** tables will be incorporated into Sec. 5.1 of the final version as supplementary experimental evidence, providing a more rigorous justification for the threshold selection.
> > >
> > > ---
> > > **2. Detailed Experimental Design for Self-Evolution**
> > >
> > > The overall framework of our self-evolution pipeline has been outlined in **Figure 1**. Below we explicitly detail the workflow:
> > > * **Data Processing**
> > >     * **Open-source Data:** Used strictly for **Stage I** of MASE. This provides the agent with the foundational capability to perform GUI tasks.
> > >     * **Simulator Data:** Used for **Stage II** of MASE. This enables the agent to achieve self-evolution by continuously interacting with the environment in a simulated setting.
> > > * **Task Instructions:**
> > >     * We utilize the canonical task templates from the **AndroidWorld** benchmark but generate **novel semantic and syntactic variations**. Specifically, we apply transformations such as changing target entities (e.g., "search for Marvel movies" $\to$ "search for Sci-Fi movies"), altering target apps, or varying the sequence of sub-goals.
> > > * **Success Determination:**
> > > Task success is determined automatically using a VLM as the Verifier. At the end of a task, we input the final screenshot and the instruction into a Verifier (in our implementation, initially it was Qwen2.5-VL-72B, and later, when the agent's performance improved, the agent performed self-assessment).
> > >     * **Criteria:** The Verifier is prompted to determine if the visual state satisfies the instruction requirements. When the agent fails to complete the full instruction, the agent employs **Hindsight Goal-Shifting** (mentioned in Sec. 5.1) to generate a new trajectory. This new trajectory will be submitted to the Verifier for judgment again until it passes the verification.
> > > * **Multi-round Evolution Setup:**
> > > The data used in each iteration is dynamically updated. The details are as follows:
> > >     * **Round $n$:** The agent performs the task and stores the corresponding trajectory.
> > >     * **Retraining:** When the threshold is reached, the model undergoes MASE training using the updated memory buffer. In our experiments, we use a mixture of 70% newly collected tasks and 30% previous data in each round to ensure continuous self-evolution.
> > >
> > > The detailed experimental design descriptions for self-evolution will be integrated into Sec. 5.1 or the Appendix of the final version to ensure complete transparency of the self-evolution process.

---

### Official Review · Reviewer_iSDf · 2026-03-03

**Soundness:** 3
**Presentation:** 3
**Significance:** 2
**Originality:** 1
**Overall Recommendation:** 3
**Confidence:** 4

**Summary:**

This paper presents SE-GA (Self-Evolving GUI Agent), a framework for autonomous GUI agents that addresses limitations in multi-step task execution. The core contributions include: (1) Test-Time Memory Extension (TTME), a hierarchical memory retrieval mechanism that provides episodic, semantic, and experiential context during inference; and (2) Memory-Augmented Self-Evolution (MASE), a training pipeline that leverages data collected by TTME to iteratively improve the agent's policy. The authors report superior performances on some of the benchmarks

**Compliance With Llm Reviewing Policy:**

Affirmed.

**Key Questions For Authors:**

What memory system is used in the "w/o TTME" configuration? What does "w/o MASE" refer to? In these configurations, are both ground training and reinforcement learning (RL) removed, or is only RL removed?

**Strengths And Weaknesses:**

**Strengths**

-   This work promptly addresses a genuine challenge in GUI agents: adapting to dynamic, long-horizon tasks.

-   The paper proposes a practical framework for solving this issue, and the performances are reasonable.


**Weaknesses**

-   While the paper mentions that this work differs from approaches using "static policies," the definition of "static" is unclear. The narrative suggests "static" refers to "static retrieval policies," but TTME seems to fall within the scope of "static" rather than "dynamic." It is important to investigate the robustness of the agent's performance based on memory. Specifically, what impact would updates to the TTME strategy have on end-to-end performance?

-   Neither component, TTME nor MASE, demonstrates a unique contribution. MASE, in particular, appears to fall entirely within the scope of RL. All RL training uses prompts, and memory augmentation is not a novel concept. TTME appears to be a relatively minor contribution based on previous agentic systems such as UI-TARS-2. Therefore, ablation studies comparing TTME with other memory systems would be beneficial.

-   Abbreviations such as MASE overlap with other concepts (e.g., Multi-Agent Self-Evolution), and the abbreviations in general (TTME, MASE, SEGA) are confusing.

---

> ### Author Rebuttal · Authors · 2026-03-30
>
> We really appreciate the reviewer for the detailed and constructive feedback. We are pleased to see that you agree with our proposed framework for addressing the long-term task problem in GUI agents. We have carefully addressed all your concerns with additional experiments and clarifications below.
>
> ---
> **Response to W(1):**
> A1: Thanks for the good question about the dynamics of TTME. In our paper, the distinction between "static" and "dynamic" is based on **whether the policy evolves with interaction experience**:
> - **Static policies** refer to policies that are trained on fixed datasets and then **frozen**, without absorbing new experiences (e.g., traditional SFT or pre-training-based agents).
> - **TTME is dynamic:** TTME is not merely a retrieval mechanism but also provides **dynamic buffer** capabilities. During reasoning, TTME continuously writes new trajectories into the experiential memory, thereby achieving real-time policy evolution, a capability lacking in static retrieval systems.
>
> Regarding the impact of TTME strategy updates on end-to-end performance, our ablation studies in Section 5.3 and Appendix C.3 indirectly validate this: Removing TTME leads to a decrease in the success rate of executing multi-step tasks, and the success rate of executing long-term tasks (30 steps or more) decreases by **20.4%**.
>
> ---
> **Response to W(2):**
> A2: We thank the reviewer for the critical assessment and the constructive suggestions. Within our framework, TTME provides high-quality data for MASE, while MASE's optimization strategy is designed to produce better data. We clarify the uniqueness of MASE and TTME below and address the request for further ablation studies.
>
> - **Uniqueness of MASE:** MASE extends beyond standard RL by introducing: 1) **Hindsight Goal-Shifting** (Eq. 17) to convert failed trajectories into valuable supervision for data efficiency; 2) **Hierarchical Rewards** (Eq. 12-15) for fine-grained spatial feedback ($R_{point}$, $R_{bbox}$). These specifically address the sparse reward and grounding precision issues inherent to GUI navigation.
> - **Uniqueness of TTME:** TTME proposes a **hierarchical memory architecture** with hybrid visual-semantic retrieval (Eq. 6), specifically suitable for the complexity of GUI spaces. Unlike UI-TARS which relies on implicit internal context, our explicit retrieval mechanism enables superior dynamic adaptation (39.0% vs 33.0% on AndroidWorld).
> - **Memory Comparison:** To address your request, **R3-Table 1** compares TTME with other memory systems (Mem0, A-MEM) with our model. TTME achieves the highest task success rate, confirming the effectiveness of our hierarchical design for GUI tasks.
> >R3-Table 1: Comparison of Memory Systems
>
> |Method&emsp;|AndroidControl-High&emsp;|&emsp;GUIOdyssey&emsp;&emsp;&emsp;|
> |---|---|---|
>
> ||Type|Grounding|SR|Type|Grounding|SR&emsp;|
> |---|---|---|---|---|---|---|
> |TTME|94.6|92.5|88.6|84.4|76.4|73.8|
> |Mem0 [1]|89.2|85.3|81.4|78.7|71.8|65.3|
> |A-MEM [2]|90.6|87.5|84.6|79.6|73.9|68.9|
>
> [1] Mem0: Building Production-Ready AI Agents with Scalable Long-Term Memory
> [2] A-MEM: Agentic Memory for LLM Agents
>
> ---
> **Response to W(3):**
> A3: Thanks for the catch. We understand that there might be some confusion regarding the abbreviations. In the revised version, we will use the full terms and provide additional semantic explanations to address this confusion.
>
> ---
> **Response to Q(1):**
> A4: We thank the reviewer for the comment. The ablation configuration in the paper is as follows:
> - **"w/o TTME":** The entire hierarchical memory module is completely removed. In this setting, the model relies solely on the base VLM's limited context window to process interaction history.
> - **"w/o MASE":** This refers to removing the **Self-Evolution Training** module (Stage II), while the **Grounding Training** (Stage I) and the **TTME** module are retained.
>
> As shown in R3-Table 2, we have updated this ablation study and conducted experiments at different stages of MASE respectively to enhance readability and verify the unique contribution of each component.
> >R3-Table 2: Extended Ablation Study
>
> |Method&emsp;&emsp;&emsp;&emsp;&emsp;|AndroidControl-High&emsp;&emsp;&emsp;|GUIOdyssey&emsp;&emsp;|
> |---|---|---|
>
> ||Type|Grounding|SR|Type|Grounding|SR &emsp;|
> |---|---|---|---|---|---|---|
> |SE-GA|94.6|92.5|88.6|84.4|76.4|73.8|
> |w/o TTME|91.9|90.8|83.0|81.0|68.4|61.4|
> |w/o MASE-Stage II|88.6|86.9|74.3|72.2|60.1|59.7|
> |**w/o MASE-Stage I**|79.0|77.2|70.6|68.7|59.3|44.0|
> |**w/o MASE**|69.7|59.1|55.7|66.4|51.8|49.3|
>
> ---
> We hope these additional experiments and clarifications adequately address your concerns. If you find the new evidence convincing, we would appreciate if you could consider updating your scores accordingly. Thank you once again for your time and effort. We look forward to incorporating these insightful discussions into the final version.

---

### Official Review · Reviewer_MpxM · 2026-03-12

**Soundness:** 3
**Presentation:** 3
**Significance:** 3
**Originality:** 3
**Overall Recommendation:** 4
**Confidence:** 3

**Summary:**

The paper proposes SE-GA which addresses  two problems in recent GUI agents methods: limited context windows and static, non-adaptive policies. It combines two components: TTME (Test-Time Memory Extension), a hierarchical retrieval system with episodic, semantic, and experiential memory; and MASE (Memory-Augmented Self-Evolution), a two-stage training pipeline using supervised fine-tuning followed by GRPO-based reinforcement learning. A novel Hindsight Goal-Shifting mechanism converts failed trajectories into useful training signal by relabeling partial completions as successful sub-goal demonstrations. SE-GA achieves  89.0% on ScreenSpot, outperforming even 72B-parameter models, 75.8% success rate on AndroidControl-High, 39.0% on AndroidWorld, well above the next-best baseline (UI-TARS 7B at 33.0%) and 83.9% step success rate on GUIOdyssey.

**Compliance With Llm Reviewing Policy:**

Affirmed.

**Final Justification:**

Thank you for the detailed rebuttal and clarifications. I will keep my original score.

**Key Questions For Authors:**

Mentioned above

**Limitations:**

Mentioned above

**Strengths And Weaknesses:**

Strengths:
- Interesting idea of incoproating a cognitive-science-inspired memory (episodic memory) into a GUI agent training pipeline
- The Hindsight Goal-Shifting idea is simple and practically valuable as it allows recycling failures rather than discarding them
- Experimental setup is thorough and strong ablations clearly isolate the contribution of each module
- SE-GA performs competitive with models 10× larger in parameter count


Weaknesses:
- An ablations that would be valuable to see is: if instead of fetching just top-k reflective summaries based on similarity if we fetch top-k trajectories from groups which have successful only, success/failure trajectories with 50:50 ratio. Because I imagine the reflectiions from past traces are usually helpful when  model is having hard time learning a task. Therefore it makes a lot more sense to find the most “informative” samples when you find the most similar samples.
- It’d also be great to have analysis on how many tokens are consumed by past trajectories and how that scales as training and evaluation progresses with SE-GA.
- Memory retrieval quality heavily depends on text and visual embedding similarity, which may be brittle for visually complex or ambiguous GUI states which require reasoning over a trajectory rather than just frames/instructions individually. Additional, discussions or analysis on how often such cases happen and examples would make paper stronger.
- Limited discussion of computational overhead introduced by hierarchical memory retrieval at inference time. How does it increase the context length? FLOPs used as number of steps scale? It’d be great if authors could add relevant analysis for that.

---

> ### Author Rebuttal · Authors · 2026-03-30
>
> We sincerely appreciate your insightful and professional feedback. We are delighted by your recognition of our core idea, particularly the cognitive-science-inspired memory design and the practical value of Hindsight Goal-Shifting. Below, we provide detailed responses to all the comments:
>
> ---
> **Response to W(1):**
> A1: This is an excellent suggestion. We believe that the direction you pointed out is of great value and is directly related to the core design concept of our TTME module. It can better clarify the essence of the memory mechanism. We have added the following ablation experiments:
> >R2-Table 1: Comparison of different retrieval strategies
>
> |Retrieval strategy|AndroidControl-High|GUIOdyssey|
> |---|---|---|
> |Top-k trajectories|75.8|83.9|
> |Mixed trajectories|76.2|82.1|
> |Success trajectories|70.0|72.5|
>
> Through the experiments, we found that both Top-k and Mixed strategies outperform Success-only retrieval, validating that **negative examples are indeed informative**. Additionally, we also discovered that the success rates between the top-k strategy and the mixed strategy were not significantly different. We speculate that this might be because the ratio of successful samples and failed samples selected by our text-image hybrid retrieval strategy was approximately 50%, so there is no need for mandatory proportion constraints. We appreciate this valuable suggestion, which led to this insightful finding.
>
> ---
> **Response to W(2):**
> A2: Thank you for your question. We provide the token consumption statistics profiled during inference: the system prompt is 3,447 tokens, and instructions are 18–102 tokens, and memory retrieval consumes 300–400 tokens per iteration.
>
> Crucially, the token count per inference step remains **relatively constant**, because although the memory repository grows continuously, each retrieval operation selects a fixed-size context (top-$k$ compressed items). Consequently, the training cost scales linearly with the dataset size rather than the memory size, ensuring that retrieval overhead remains bounded as the agent accumulates more experience.
>
> ---
> **Response to W(3):**
> A3: Thank you for your question. This is indeed a concern about the reliability of our retrieval mechanism. We employ multiple mechanisms to handle visual complexity and ambiguity in GUI states. Firstly, we use an embedding methods combining textual and visual features, which can capture semantic and visual similarities, emphasizing overall task matching degree rather than just text matching or visual matching. Secondly, our hierarchical retrieval strategy records recent action background information, which can better help the agent understand the current complex GUI state.
>
> We illustrate this with an example. Consider a scenario where the agent encounters a visually ambiguous interface state: the screen shows a text editor with an open document, making it unclear whether this is "composing a new email" or "editing an existing draft" based solely on the current screenshot (visual similarity to both tasks is high).
>
> In this case, relying only on visual-text embedding similarity would retrieve irrelevant memories (e.g., a "edit document" memory with 0.85 similarity vs. a "compose email" memory with 0.82). However, our **hierarchical retrieval** leverages the **episodic trajectory context**: the preceding actions in the current episode indicated the agent just clicked "New Message" (stored in the short-term episodic buffer). This trajectory information disambiguates the current state, ensuring that retrieval prioritizes "email composition" memories, despite the single-frame visual ambiguity.
>
> ---
> **Response to W(4):**
> A4: Thank you for the constructive feedback. SE-GA is designed to **maintain constant computational** overhead regardless of total task steps, avoiding unbounded scaling:
> - **Context Length**: We prevent linear context growth through two mechanisms:
>     - **Episodic Memory**: Uses a sliding window with a fixed horizon $H$ (Eq. 2), discarding stale history.
>     - **Semantic/Experiential Memory**: Retrieves only Top-$K$ entries (Sec. 4.1.2-3).
>     Consequently, the input context is bounded. As shown in Appendix B.2, we enforce a maximum prompt length of 6144 tokens.
> - **FLOPs Scaling**: Since the input context length per step is bounded by design ($H + K \cdot L_{entry}$), the FLOPs per VLM forward pass remain constant ($O(1)$) relative to the total number of steps $t$. The retrieval cost is negligible compared to the VLM inference. This ensures efficient long-horizon execution.
>
> ---
> We appreciate your recognition of this work. We hope our responses have effectively addressed your concerns and we are more than happy to include all these discussions in the camera-ready version of this work.

---

> > ### Author Rebuttal · Reviewer_MpxM · 2026-04-06
> >
> > NA

---

### Official Review · Reviewer_cGKM · 2026-03-13

**Soundness:** 3
**Presentation:** 3
**Significance:** 3
**Originality:** 3
**Overall Recommendation:** 5
**Confidence:** 2

**Summary:**

This paper presents SE-GA, a novel framework that leverages structured memory for GUI navigation both at inference and for agent policy self-improvement. SE-GA is split into two components, TTME, a hierarchical memory mechanism that stores episodic, semantic, and experiential memory as context during inference, and MASE, a training pipeline for GUI action grounding and self-evolution. Furthermore, the authors introduce a "Hindsight Goal-Shifting Mechanism", where failed trajectories with successful sub-goals are relabeled to be successful instances of the subgoal. SE-GA (using a base Qwen2.5-VL-7B model) is evaluated against fifteen recent baselines comprising closed-source VLMs, open-source VLMs, and specialized GUI-agents on three offline and one online GUI benchmarks: ScreenSpot, AndroidControl, GUIOdyssey, and AndroidWorld. SE-GA outperforms baselines on several benchmarks (89% on ScreenSpot, 39% on AndroidWorld).

**Compliance With Llm Reviewing Policy:**

Affirmed.

**Final Justification:**

My final justification is to give this paper an accept.
I found the paper to be strong but had some weaknesses in regards to presentation and experiments. The authors' detailed rebuttals and the additional ablation experiments have sufficiently addressed my concerns.

**Key Questions For Authors:**

1. How many rounds of MASE self-evolution training were run to produce the reported results? Was the evaluation performance measured after one or after multiple iterative rounds? The paper claims that SE-GA enables continuous self-evolution, but it is not clear whether the experimental setup tries to demonstrate this.
2. Which specific baseline results were taken from UI-TARS's reported numbers instead of evaluating in the same experimental conditions? For those results, do the authors believe the evaluation protocols are sufficient for a fair comparison? The use of external benchmark numbers from the UI-TARS paper is briefly mentioned, but does not go into detail why nor mention which baselines model numbers were used. This information would be helpful for judging Soundness in the paper evaluation.
3. Are there more detailed statistics of the 4K trajectory training dataset that was collected (i.e. average number of steps per trajectory, total number of screens, distribution of trajectory lengths...etc.)? These details would with judging the Soundness of the training setup.

**Limitations:**

No. While the authors provide a detailed impact statement, there is no Limitations section with mention of methodological limitations or potential future directions. Some examples of methodological limitations include:
* Reliance on external results from UI-TARS (Wang et al. 2025) may cause comparison validity concerns.
* MASE seems to have only been run for one round, indicating effects of long-term self-evolution have not been tested.

**Strengths And Weaknesses:**

# Soundness
The authors conduct thorough experiments to evaluate the performance of SE-GA against multiple different GUI agent baselines and on various GUI evaluation benchmarks. The authors also helpfully report detailed training hyperparameters and GPU details. The ablation study measuring task success rate of SE-GA without TTME or MASE for short and long horizon tasks shows how both mechanisms contribute to the framework. However, the self-evolution training seems to only be run for one round, which shows promising results, but reporting results after evaluating over multiple iterative rounds would better strengthen the self-evolution claim.

# Presentation
The paper formalizes the problem definition well and clearly explains the different details regarding the TTME and MASE components. Detailed training implementation information is provided in the appendix. However, the case study figures (Appendix C.1 and C.2) are not easily interpretable. The images are appended together with no clear direction and it is not immediately clear why a trajectory fails or succeeds just viewing the images. For the ablation figure in Appendix C.3, it is not clear from which benchmark evaluation the trajectories and success rates are derived. A formatting nitpick I found is that there are duplicate citations in the References (see SeeClick, OS-atlas, Aguvis). Also, the authors describe the semantic, experiential, and episodic memory types, but I think the paper would benefit if concrete examples were displayed somewhere like the appendix.

# Significance
Write: The findings in this paper are significant. We see that SE-GA using a 7B Qwen base model in some evaluations surpasses larger models like UI-TARS 72B. This suggests that memory architecture and effective training pipeline are as important as model scale, which has implications for future GUI agent design. Furthermore, SE-GA maintains robust performance on long horizon tasks relative to baseline GUI agents. While SE-GA outperforms other model baselines on AndroidWorld, its 39% success rate indicates there is significant room for improvement.

# Originality
Write: The authors' methods are valuable extensions of prior works. Their proposal of hindsight goal shifting by turning failed trajectories into valid sub-goal task trajectories to gather training data is a great application of hindsight experience replay for GUI task labelling (Andrychowicz et al. 2018). Moreover, the use of different types of episodic, semantic, and experiential memories as retrieval context is a nice extension of previous works leveraging memory structures for GUI agents like Show UI (Lin et al. 2025).

# References
* Andrychowicz, M., Wolski, F., Ray, A., Schneider, J., Fong, R., Welinder, P., McGrew, B., Tobin, J., Abbeel, P., and Zaremba, W. Hindsight experience replay, 2018. URL https://arxiv.org/abs/1707.01495.
* Lin, K. Q., Li, L., Gao, D., Yang, Z., Wu, S., Bai, Z., Lei, S. W., Wang, L., and Shou, M. Z. Showui: One vision-language-action model for gui visual agent. In  Proceedings of the Computer Vision and Pattern Recognition Conference, pp. 19498–19508, 2025.
* Wang, H., Zou, H., Song, H., Feng, J., Fang, J., Lu, J., Liu, L., Luo, Q., Liang, S., Huang, S., et al. Ui-tars2 technical report: Advancing gui agent with multi-turn  reinforcement learning. arXiv preprint arXiv:2509.02544,  2025a.

---

> ### Author Rebuttal · Authors · 2026-03-30
>
> We really appreciate the reviewer cGKM for the detailed and constructive feedback. We are pleased to see that you found our proposed framework for leveraging structured memory to perform GUI navigation task, both in inference and for agent policy self-improvement, to be a meaningful and practically impactful direction. Below, we provide detailed responses to all the comments:
>
> ---
> **Response to Q(1):**
> A1: Thanks for the good question. We also believe that demonstrating the iterative improvement process over multiple rounds is crucial for verifying the self-evolution capability. Actually, MASE self-evolution training was run twice in this paper, and the performance was evaluated after **two rounds of iterations**. To prevent data leakage, data collection was conducted within a simulator. R1-Table 1 shows the results of different self-evolution rounds, which indicates that our method can achieve continuous self-evolution:
> >R1-Table 1: Comparison of different rounds
>
> |Benchmark|Round 1|Round 2|Round 3|
> |---|---|---|---|
> |ScreenSpot|79.3|86.0|89.0|
> |AndroidControl-Low|68.3|75.5|88.6|
> |AndroidControl-High|55.9|71.3|75.8|
> |GUIOdyssey|52.3|75.1|83.9|
> |AndroidWorld|28.6|34.5|39.0|
>
> ---
> **Response to Q(2):**
> A2: As you've pointed out, the paper does not provide detailed information about the baseline results, which undermines the rigor of the paper's assessment. Therefore, we are sharing more experimental details here. The baseline results from the UI-TARS paper are: **UI-TARS, GPT-4o, Gemini, Claude3, InternVL, Aria-UI, Aguvis, UGround, SeeClick**.
> **Justification for Using External Results:**
> - **Reproducibility**: Some baselines use proprietary training data and infrastructure that cannot be fully reproduced. Using their reported numbers ensures fair comparisons against their official benchmarks.
> - **Benchmark Consistency**: All the evaluations in this paper were performed on the same benchmark versions and according to the same evaluation procedures, thus enabling direct and reasonable comparisons.
>
> ---
> **Response to Q(3):**
> A3: We thank the reviewer for the comment. Regarding the dataset we have collected, we plan to make it open-source in the near future. Here, we will share with the reviewers **more detailed statistical information** of our dataset, so that reviewers can assess the rigor of the training settings. The dataset exhibits a broad distribution of task lengths (1–31 steps) with a median of 12 steps, providing diverse and realistic training scenarios for GUI agents:
> >R1-Table 2: The detailed information of the dataset
>
> |Metric|Value|
> |---|---|
> |Total trajectories|4,007|
> |Total screens|4,007|
> |Average steps per trajectory|11.89|
> |Minimum trajectory length|1|
> |Maximum trajectory length|31|
> |Median trajectory length|12|
> |Short Task (1-7 steps)|67% (444)|
> |Medium Task (8-15 steps)|22% (145)|
> |Long Task (16-47 steps)|11% (71)|
>
> We are committed to open science and will release our code, trained models, and dataset upon acceptance. We welcome any additional feedback or questions.
>
> ---
> **Response to Limitations:**
> A4: For the detailed discussion on MASE and the baseline, we provided the answers in A1 and A2. Additionally, regarding the limitations of the methodology or potential future directions, we will provide detailed explanations in the final version, the following are the simplified version:
>
> The methodological limitation of this study is Memory Retrieval Efficiency. To address these limitations and further advance GUI agents, we identify three key directions for future research. (1) Scaling to larger training datasets with diverse task types. (2) Exploring hierarchical task decomposition for long-horizon planning. (3) Investigating transfer learning across different GUI platforms.
>
> ---
> **Response to Presentation:**
> A5: We are aware that there are issues regarding the clarity of the charts in our case study (Appendix C.1 and C.2). We plan to completely redesign the visual content of the case study and add explanations to clarify all the important visual elements. We will update Appendix C.3 to clearly specify the benchmark source is GUIOdyssey. Otherwise, thank you for pointing out the situation of duplicate citations in our reference section. We will remove all duplicate citations and cross-checked the entire reference list to ensure no other duplicates exist.
>
> We also agree that concrete examples would greatly enhance the understanding. We have added some comprehensive examples in Appendix C, including detailed examples of each memory type, illustrating how they are retrieved and used during inference.
>
> ---
> We are happy to see from your comments that you have an overall positive view of our work, and we respectfully ask if you'd be comfortable to adjust your ratings, based on all the new experiments and justification to your questions. Thank you once again for your time and effort. We look forward to incorporating these insightful discussions into the final camera-ready version.

---

> > ### Author Rebuttal · Reviewer_cGKM · 2026-04-04
> >
> > Thank you for the detailed rebuttal. My questions have been sufficiently answered, and it seems additional ablations have been added. The self-evolution table is helpful, but it would be interesting to see the trend of improvement over a longer period of time (e.g., 10 rounds). The benchmarks justification is reasonable. This may be a clerical error, I see that your total screens and table trajectories in R1-Table2 are the same number. I will change my score accordingly, and I look forward to seeing the changes in the final version.

---

> > > ### Author Response · Authors · 2026-04-04
> > >
> > > We are very pleased to have addressed your concerns and thank you very much for raising the score! All the revisions and improvements will be incorporated into the final version. We will carefully review all the details and look forward to presenting them.

---

### Decision · Program_Chairs · 2026-04-30

**Decision:**

Accept (regular)

**Comment:**

This paper introduces a method called Self-Evolving GUI Agent (SE-GA), designed to address two issues with recent GUI agent techniques: limited context windows and static, non-adaptive policies that cannot adapt to dynamic environments. SE-GA leverages two main components. One enables long-term planning through a hierarchical system that dynamically retrieves semantic context during inference. The other implements a two-stage training procedure for GUI action grounding and self-evolution, and is designed to support continuous learning.

Overall, reviewers shared a positive impression of this work's contributions and significance. Two reviewers highlighted the thorough experiments conducted by the authors in comparing SE-GA against multiple relevant baselines across various benchmark problems. They noted, in particular, the strong empirical results, with the method often outperforming much larger models, sometimes with roughly 10x more parameters. These results help support the authors' claims and validate the method's effectiveness, especially on long-horizon tasks. One reviewer also acknowledged the well-designed ablation studies, which they argued convincingly show why the two proposed mechanisms underlying SE-GA are needed and how they contribute to the framework's performance. Finally, one reviewer noted that SE-GA's ability to outperform baselines with an order of magnitude more parameters "*suggests that memory architecture and effective training pipeline are as important as model scale, which has implications for future GUI agent design*".

During the discussion, a few concerns were raised. One reviewer pointed out that the paper included limited discussion of the computational overhead introduced by SE-GA's hierarchical memory retrieval mechanism. They were also concerned with the number of tokens consumed after analyzing previous trajectories and how this scales as training and evaluation progress. Post-rebuttal, in light of the authors' responses, that reviewer stated that all of their questions and concerns had been fully resolved. Another reviewer had a few technical questions, most of which were adequately addressed in the rebuttal, but noted that one key concern remained. In particular, they felt that the proposed retraining trigger, although now better supported by additional analysis introduced during the discussion phase, appeared to be a system-specific heuristic rather than a sufficiently justified general principle. Finally, one reviewer pointed out presentation issues, such as duplicate citations and some images that, in their current form, are difficult to interpret.

All reviewers agreed that this paper addresses an important problem, is well written and clear, and presents strong experimental results that are significant and help support the authors' main claims. They also highlighted the well-designed comparisons and ablation studies, while also identifying a few technical concerns and points that could be clarified in the manuscript. They encouraged the authors to further update the paper to address the points raised in the reviews, incorporate the new experimental results presented during the discussion phase, and clarify the main questions raised during the discussion. All reviewers agreed that addressing these points would substantially strengthen the paper and help highlight its important contributions.